# With Limited Data for Multimodal Alignment, Let the STRUCTURE Guide You

**Fabian Gröger**[1,2,3,*], **Shuo Wen**[1,*], **Huyen Le**[1], **Maria Brbić**[1]

[1]EPFL   [2]University of Basel   [3]HSLU

`brbiclab.epfl.ch/projects/structure`

## Abstract

Multimodal models have demonstrated powerful capabilities in complex tasks requiring multimodal alignment, including zero-shot classification and cross-modal retrieval. However, existing models typically rely on millions of paired multimodal samples, which are prohibitively expensive or infeasible to obtain in many domains. In this work, we explore the feasibility of building multimodal models with limited amount of paired data by aligning pretrained unimodal foundation models. We show that high-quality alignment is possible with as few as tens of thousands of paired samples—less than $1\%$ of the data typically used in the field. To achieve this, we introduce STRUCTURE, an effective regularization technique that preserves the neighborhood geometry of the latent space of unimodal encoders. Additionally, we show that aligning last layers is often suboptimal and demonstrate the benefits of aligning the layers with the highest representational similarity across modalities. These two components can be readily incorporated into existing alignment methods, yielding substantial gains across 24 zero-shot image classification and retrieval benchmarks, with average relative improvement of $51.6\%$ in classification and $91.8\%$ in retrieval tasks. Our results highlight the effectiveness and broad applicability of our framework for limited-sample multimodal learning and offer a promising path forward for resource-constrained domains.

## 1 Introduction

Unimodal foundation models (FMs) have achieved remarkable progress in recent years, demonstrating high performance on a variety of complex tasks across diverse domains. Large language models (LLMs) [1, 2] have shown unparalleled abilities across a wide range of natural language understanding and generation tasks, reaching human-level performance on many complex reasoning and comprehension tasks. In parallel, vision FMs [3, 4] have delivered similar breakthroughs in visual perception, enabling systems to perform tasks such as object recognition and segmentation with minimal supervision. In scientific domains, models such as AlphaFold [5] and ESM2 [6] have demonstrated the potential of large-scale models to solve highly specialized and complex problems, including accurate protein structure prediction and sequence-based modeling of biomolecular functions.

While these models are highly effective within their respective modalities, many applications involve multimodal data, where it is important to map different modalities into a shared representation space to enable meaningful cross-modal comparison, retrieval, and classification. Pioneering multimodal models like CLIP [7] and CLAP [8] align visual and auditory inputs with language through contrastive learning and achieve strong zero-shot performance across a broad spectrum of multimodal tasks. However, these powerful multimodal models rely on vast amounts of paired training data (*e.g.*, CLIP uses 400M pairs), which are often unavailable in many domains like healthcare and biology, where collecting high-quality multimodal data is expensive and labor-intensive.

---

[*]Joint first authorship

39th Conference on Neural Information Processing Systems (NeurIPS 2025).

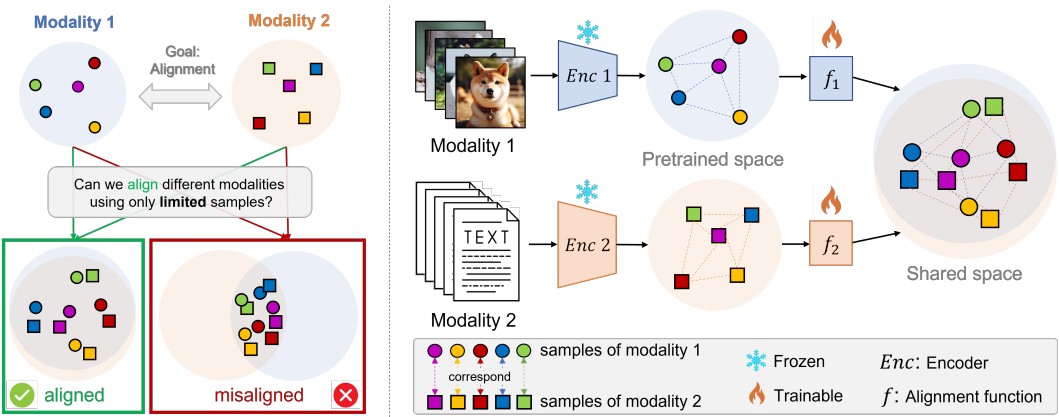

Figure 1: Overview of the proposed approach for cross-modal alignment with limited data. The objective is to align representations from two modalities (*e.g.*, images and text) into a shared embedding space. The central challenge is guiding the model toward a well-aligned solution, rather than a misaligned one, when only a small amount of paired data is available. The key idea is to freeze pretrained encoders and learn lightweight alignment functions that preserve each modality's pretrained latent structure during alignment.

A promising direction to address this limitation is to leverage powerful representations from pretrained unimodal FMs for multimodal alignment. However, existing alignment methods [9–11] still require large quantities of paired examples, often tens of millions, to learn a shared embedding space effectively. On the other hand, unsupervised techniques [12, 13] do not use labeled data but primarily perform sample-level matching, thus failing to construct a shared embedding space. This opens a fundamental question: *Can we align FMs from different modalities into a shared representation space using a limited number of multimodal samples?*

In this work, we answer this question affirmatively by introducing a modular alignment strategy that requires as few as tens of thousands of paired examples, less than $1\%$ of the data used by existing modalities alignment methods [9–11]. Specifically, we introduce STRUCTURE, a novel regularization technique designed to preserve the multiscale neighborhood structure of each unimodal latent space during alignment, ensuring that the relationships between samples captured by the unimodal encoders are retained. In addition, based on the observation that higher similarity correlates with better alignment performance, we propose to align those layers with the highest representational similarity. Incorporating both components into existing alignment methods leads to consistent relative performance gains across 22 zero-shot classification and two retrieval benchmarks, averaging $51.6\%$ in classification and $91.8\%$ in retrieval, demonstrating superior performance in low-data settings. Finally, we show that incorporating just a few labeled examples per class from the target domain into the training set can effectively bridge the performance gap to large-scale multimodal models trained on hundreds of millions of multimodal data, highlighting the importance of domain coverage over sheer data volume in multimodal alignment.

## 2 Related work

**Multimodal foundation models.** Recent advances in multimodal FMs have produced unified architectures capable of understanding and generating content across vision and language. Early models such as CLIP [7] and ALIGN [14] use contrastive learning to align image and text embeddings, achieving strong zero-shot performance in classification and retrieval. In contrast, models like BLIP [15] and GIT [16] adopt generative pretraining to support tasks such as image captioning and visual question answering. More recent efforts, including FLAVA [17], PaLI [18], Kosmos-1 [19], and Gemini [20], aim for broad task coverage through unified multimodal training, reflecting a shift toward scalable, general-purpose multimodal systems. Despite their impressive capabilities, existing multimodal models rely heavily on massive paired datasets, often hundreds of millions of samples. In contrast, in this work we explore how to align powerful unimodal models that are readily available in many domains using only limited multimodal supervision, reducing reliance on large-scale paired data and enabling multimodal learning in low-data regimes.

**Parameter-frozen modality alignment.** Parameter-frozen modality alignment is an efficient strategy for building multimodal systems by aligning pretrained unimodal encoders without updating their internal weights. *Unsupervised alignment methods* seek to align representations without relying on paired data. Techniques such as Centered Kernel Alignment [12] focus on maximizing representational similarity between modalities, leveraging shared structural patterns. However, a key limitation of these methods [13, 12, 21] is that they primarily perform sample-level matching, thus failing to construct a shared embedding space. Furthermore, they are not able to utilize paired data even when available, missing out on rich alignment signals that could significantly enhance performance. In contrast, *supervised alignment methods* assume access to abundant paired data, such as millions of paired samples, and train lightweight alignment modules (*e.g.*, linear projections or MLPs) while keeping the encoders frozen. These approaches [9, 10] have achieved strong performance in tasks like zero-shot classification and retrieval. However, these methods are limited by their dependence on large-scale multimodal datasets, requiring tens of millions of paired samples to effectively train the models. In contrast, our work explores parameter-frozen alignment in a low-data regime, with tens of thousands of paired samples, which is $0.02\%$ of the paired samples used by previous works. To address the challenges of low-data regimes, we propose an effective strategy that can be easily integrated into existing methods [9, 10] to make them more effective under low-data conditions.

Other works have also focused on data-efficient alignment. FuseMix [22] shares our motivation of using limited paired data but focuses on improving training efficiency via latent space augmentations. Our geometric regularization and layer selection are complementary to their approach, and as shown in our experiments, combining them yields further gains. Learning-free methods like ASIF [23] also leverage neighborhood structure, but for direct matching without training an alignment function.

**Platonic representation hypothesis.** The Platonic representation hypothesis [24] suggests that models from different modalities can converge to similar internal representations. This insight motivates the possibility of aligning separate representation spaces. While earlier multimodal FMs like CLIP [7] depend on end-to-end training with vast amounts of paired data, the Platonic view offers a more efficient alternative: aligning independently trained unimodal models within a shared embedding space. This approach opens the door to scalable and flexible multimodal learning without the need for joint training, and it has recently spurred interest in supervised and unsupervised alignment techniques [9, 12]. Building on this direction, which is related to the broader field of geometric priors in representation alignment [25–28], our work explores its potential in the challenging low-data setting.

## 3 Problem setup

We consider the task of aligning representations from independently pretrained encoders across different modalities. As illustrated in Figure 1, we keep the encoders frozen and learn lightweight alignment functions that map from each modality's latent space into a shared space where semantically related samples are close. We next formalize this setup.

**Modality alignment with pretrained representations.** Let $\mathcal{X}_1 \subseteq \mathbb{R}^{d_1}$ and $\mathcal{X}_2 \subseteq \mathbb{R}^{d_2}$ be the latent spaces of two pretrained unimodal encoders, which correspond to outputs from either *the last or intermediate layers* of the encoders, and $d_1$ and $d_2$ be their respective dimensions, which do not need to be equal. Given a set of $N$ paired multimodal samples $\mathcal{D} = \{(x_1^i, x_2^i)\}_{i=1}^N$, where $x_1^i \in \mathcal{X}_1$ and $x_2^i \in \mathcal{X}_2$, our goal is to learn two alignment functions $f_1 : \mathbb{R}^{d_1} \to \mathcal{A}$ and $f_2 : \mathbb{R}^{d_2} \to \mathcal{A}$, which map the modality-specific spaces $\mathcal{X}_1$ and $\mathcal{X}_2$ into a shared embedding space $\mathcal{A} \subseteq \mathbb{R}^k$ of dimension $k$. Alignment in the shared space is achieved when the paired samples $(x_1^i, x_2^i)$ are closer to each other than to any non-paired sample. Specifically,

$$\text{sim}(f_1(x_1^i), f_2(x_2^i)) \geq \text{sim}(f_1(x_1^i), f_2(x_2^j)) \quad \forall j \neq i, \tag{1}$$

where $\text{sim}(\cdot)$ denotes a similarity function, such as cosine similarity.

Different from previous works, we focus on this alignment problem under the challenging condition of *limited paired data*. Specifically, we focus on the scenario when $N$ is relatively small (*i.e.*, tens of thousands of samples) compared to tens of millions of paired samples considered in previous works [9, 10].

We unify different alignment methods under a joint framework that consists of three main components: *(i)* modality-specific latent spaces $\mathcal{X}_1$ and $\mathcal{X}_2$, *(ii)* alignment functions $f_1$ and $f_2$, and *(iii)* the objective

Table 1: Overview of existing methods and our approach within the modality alignment framework. $\mathcal{L}_C$ represents the standard symmetric contrastive losses from CLIP [7]. $\mathcal{R}_S$ denotes STRUCTURE, a regularization term proposed in our work.

| Method | Modality-specific spaces | Alignment functions $f$ | Objective function $\mathcal{L}_A$ |
|---|---|---|---|
| Linear mapping [10] | Last layers | Linear | $\mathcal{L}_C$ |
| Non-linear mapping [9] | Last layers | MLP | $\mathcal{L}_C$ |
| Token alignment [9] | Last layers | Token level MLP | $\mathcal{L}_C$ |
| CSA [29] | Last layers | Linear | Reformulated $\mathcal{L}_C$ |
| Our work | Most similar layers | Any | Any + $\mathcal{R}_S$ |

function $\mathcal{L}_A$, which guides the construction of the shared space $\mathcal{A}$. Table 1 summarizes how existing methods instantiate each of these components. In our work, we propose a general framework that can work with any alignment function and any objective function by regularizing the objective using the STRUCTURE regularization $\mathcal{R}_S$, and using as modality-specific latent layers those layers that have highest representational similarity. We introduce both components in the following section.

## 4    Multimodal alignment with limited data

To align pretrained unimodal encoders using a limited number of paired multimodal samples, we propose two key components: *(i)* STRUCTURE, a regularization that preserves the intrinsic geometry of each modality's latent space, and *(ii)* a strategy for selecting modality-specific latent space, those layer pairs that have highest representational similarity. Both components can be seamlessly integrated into existing alignment methods [9, 10, 29] (see Table 1).

**Preserving neighborhood via STRUCTURE regularization.**    With a limited number of paired samples, it is crucial to preserve the latent structure of pretrained unimodal encoders, trained on millions or even billions of examples, as they encode meaningful relationships between samples. To achieve this, we introduce STRUCTURE, a regularization term that preserves the neighborhood relationships of the pretrained space of unimodal encoders within the aligned space.

Given a modality-specific space $\mathcal{X} \subseteq \mathbb{R}^d$ and its corresponding shared space $\mathcal{A} \subseteq \mathbb{R}^k$, where $\mathcal{X}$ can be either $\mathcal{X}_1$ or $\mathcal{X}_2$, the regularization term aims to ensure the hierarchical (*i.e.*, multi-scale) consistency between the relationships expressed by $\mathcal{X}$ and $\mathcal{A}$. Specifically, each sample $x_i \in \mathcal{X}$ and $a_i \in \mathcal{A}$ (*i.e.*, $a_i = f(x_i)$) is first $\ell_2$-normalised, $\widehat{x}_i = x_i/\|x_i\|_2, \widehat{a}_i = a_i/\|a_i\|_2$, and then centered to remove any global translation bias:

$$\widetilde{x}_i = \widehat{x}_i - \tfrac{1}{N} \sum_{j=1}^{N} \widehat{x}_j, \qquad \widetilde{a}_i = \widehat{a}_i - \tfrac{1}{N} \sum_{j=1}^{N} \widehat{a}_j. \tag{2}$$

We denote the normalized and centered matrices as $\widetilde{X} = [\widetilde{x}_1, \widetilde{x}_2, \cdots, \widetilde{x}_N] \in \mathbb{R}^{N \times d}$ and $\widetilde{A} = [\widetilde{a}_1, \widetilde{a}_2, \cdots, \widetilde{a}_N] \in \mathbb{R}^{N \times k}$.

The (scaled) similarity matrices are computed with temperature $\tau > 0$ as:

$$S_X = \frac{\widetilde{X}\widetilde{X}^\top}{\tau}, \quad S_A = \frac{\widetilde{A}\widetilde{A}^\top}{\tau}. \tag{3}$$

To interpret the similarities as probability distributions, we apply a softmax function row-wise. Specifically, we define $P_X = \mathrm{softmax}(S_X)$, $P_A = \mathrm{softmax}(S_A)$ as the corresponding matrices with the similarity distributions of the respective spaces.

To capture relationships reachable by exactly $l$ hops on the similarity graph, we define for each hierarchical level $l = 1, \ldots, L$, where $L \in \mathbb{N}$ is the total number of levels[2]:

$$P_X^{(l)} = (P_X)^l, \quad P_A^{(l)} = (P_A)^l. \tag{4}$$

Intuitively, $P_X$ can be seen as the transition matrix of a random walk on the data manifold. The $l$-hop matrices thus capture multi-step relationships and progressively more global structure.

---

[2]For a square matrix $P$, the power $P^l$ is defined by repeated matrix multiplication, *i.e.*, $P^l = P\,P\cdots P$ (with $l$ factors), and should not be confused with entrywise exponentiation, generally $(P^l)_{ij} \neq P_{ij}^l$.

The key idea of our regularization is to enforce consistency between the structural relationships, *i.e.*, the relative positions and neighborhood structure in the embedding space, captured by $\mathcal{X}$ and $\mathcal{A}$. Thus, we employ Jensen-Shannon divergence because of its symmetric nature to measure the divergence between similarity distributions. Specifically, at each level $l$, we define the level-specific divergence as:

$$\text{JS}(P_X^{(l)}, P_A^{(l)}) = \frac{1}{2} D_{\text{KL}}\left(P_X^{(l)} \middle\| M^{(l)}\right) + \frac{1}{2} D_{\text{KL}}\left(P_A^{(l)} \middle\| M^{(l)}\right), \tag{5}$$

where

$$M^{(l)} = \frac{1}{2}\left(P_X^{(l)} + P_A^{(l)}\right), \tag{6}$$

and $D_{\text{KL}}$ is the Kullback-Leibler divergence. In practice, a small constant $\varepsilon$ is added inside the logarithm for numerical stability, *i.e.*, $P_X^{(l)} + \varepsilon$ and $P_A^{(l)} + \varepsilon$.

The final formulation of the STRUCTURE regularizer is a weighted average of the divergences across levels, where lower levels are weighted more heavily to counteract the more concentrated distributions of the higher levels:

$$\mathcal{R}_{\text{S}}^{(L)}(X, A) = \frac{1}{L} \sum_{l=1}^{L} \frac{\text{JS}(P_X^{(l)}, P_A^{(l)})}{l}. \tag{7}$$

We denote $\mathcal{R}_{\text{S}}^{(L)}$ as the regularization that operates on $L$ levels, and set it to 1 if not otherwise specified.

Together with any objective function $\mathcal{L}_A$ used for representation alignment (e.g., $\mathcal{L}_C$ in work [9, 10]), the combined loss is defined as

$$\mathcal{L} = \mathcal{L}_{\text{A}} + \lambda\big(\underbrace{\mathcal{R}_{\text{S}}^{(L)}(X_1, f_1(X_1))}_{\text{Reg. for Modality 1}} + \underbrace{\mathcal{R}_{\text{S}}^{(L)}(X_2, f_2(X_2))}_{\text{Reg. for Modality 2}}\big), \tag{8}$$

where $\lambda$ is the regularization weight.

Each $\text{JS}(\cdot)$ is non-negative and vanishes *iff* its inputs coincide, with equality exactly when $P_X = P_A$ for all $l$, resulting from perfect multi-scale alignment. Moreover, by virtue $\mathcal{R}_{\text{S}}$ is invariant to global scaling, translations, and orthonormal rotations, depending solely on the intrinsic, hierarchical relational structure of the embeddings. Since $\mathcal{R}_{\text{S}}$ operates in sample space, and thus is influenced by the size of $N$, we investigate the generalization gap between the empirical and expected values of the regularization.

**Lemma 1** (Generalization bound). *The generalization gap between the empirical and the expected values of $\mathcal{R}_{\text{S}}$ is bounded by*

$$\left|\hat{\mathcal{R}}_N - \mathcal{R}^\star\right| \leq \mathcal{O}\left(\frac{1}{\sqrt{N}}\right). \tag{9}$$

*where $\hat{\mathcal{R}}_N$ is the empirical STRUCTURE regularizer and $\mathcal{R}^\star$ is its expectation over the data distribution. This shows that the empirical regularizer faithfully approximates its population expectation as the number of samples increases.*

*Proof.* Given in Appendix C.

**Similarity-based layer selection.** In parameter-frozen alignment, the quality of alignment is closely linked to the representational similarity between the unimodal representation spaces $\mathcal{X}_1$ and $\mathcal{X}_2$. Given two unimodal FMs, these spaces typically correspond to different layers of the models. Therefore, selecting the appropriate layers for alignment is critical. To identify the layers with the greatest potential for effective alignment, a metric is required to quantify the similarity between these representation spaces. Examples include Centered Kernel Alignment (CKA) [30], which measures the overall correspondence of pairwise relationships by comparing centered Gram matrices, unbiased CKA [31], which further removes the systematic overestimation inherent in standard CKA statistics when sample sizes are small, and mutual $k$-nearest-neighbor (kNN) [24], which identifies the $k$ closest samples in each modality's feature space and records the fraction of neighbors they share. Prior work has, however, solely relied on aligning models at their last layers [9], ignoring layer-based similarity. We instead challenge this approach and show in Figure 2 a strong correlation between the representational

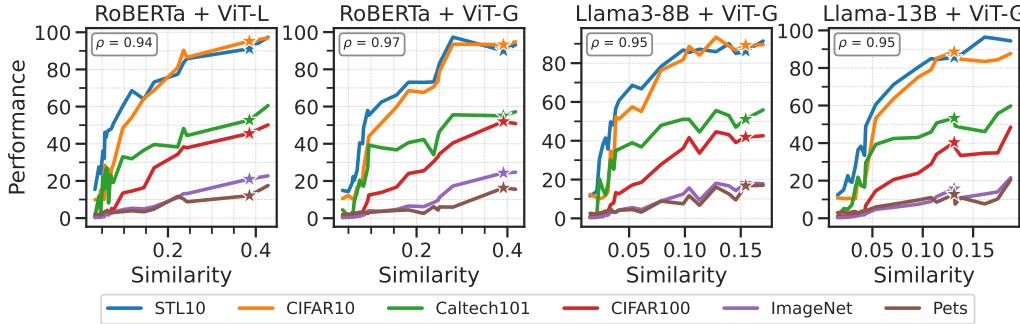

Figure 2: Zero-shot performance of different model combinations when aligning different layers as a function of their representational similarity measured in mutual kNN (MkNN). Here, the star indicates the performance achieved when aligning the last layers of the models, and $\rho$ is the average Spearman's rank correlation coefficient across different datasets.

similarity measured in terms of mutual kNN of the layers of pretrained unimodal encoders and downstream zero-shot performance once aligned using a single linear layer with our regularization.

We thus propose the following procedure for layer selection: *(i)* compute the representational similarity between all pairs of layers in terms of mutual kNN on a small set of paired samples (in the order of 5,000 pairs), typically randomly selected from the training set, and *(ii)* choose the layers with the highest similarity for alignment. Throughout our work, we compute representational similarity according to mutual kNN with $k$ chosen according to Rice's criterion[3], similar to what has been used in prior work [24] to measure similarities of different latent spaces. We show that this selection procedure leads to consistent results with different subset sizes and repeats in Appendix H.2.

## 5   Experiments

**Experimental setup.**   In our experiments, we align frozen pretrained unimodal encoders using a limited number of image–text pairs. To align models, we use the MS COCO train split consisting of 80,000 paired samples [33]. To demonstrate versatility of the STRUCTURE regularization, we apply it to different alignment strategies shown in Table 1, including linear mapping (Linear) [10], non-linear mapping (MLP) [9], and CSA [29]. In addition, to evaluate the benefits of selecting layers based on the similarity of representations rather than the last layer, we trained the models on either the final layers ("*Last*") or the similarity-based selected layers ("*Similar*"). Experiments are performed with different combinations of models. By default, we consider RoBERTa [34] and DINOv2 ViT Giant [3] as our unimodal encoders.

We evaluate model performance on zero-shot classification and cross-modal retrieval tasks. For zero-shot classification, we consider 22 datasets from the CLIP benchmark [7]. For cross-modal retrieval, we consider the Flickr30 and MS COCO test splits to evaluate both text-to-image and image-to-text performance. Detailed dataset descriptions and training configurations are described in Appendices D and E, respectively.

**Performance comparison.**   We conduct a comprehensive evaluation of zero-shot classification and cross-modal retrieval performance across diverse datasets and alignment techniques. Table 2 shows that adding STRUCTURE regularization helps improve generalization performance across all three alignment techniques. On average, STRUCTURE yields substantial relative gains in both zero-shot classification and cross-modal retrieval. Specifically, for zero-shot classification, the average relative improvement from using STRUCTURE (compared to the baseline using the most similar layers without regularization) is 74.0% for MLP, 68.4% for Linear, and 26.8% for CSA. In retrieval tasks, the corresponding relative improvements are 137.0%, 122.8%, and 15.9%, respectively. These results highlight that MLP and Linear alignment strategies benefit the most from STRUCTURE, while CSA shows more moderate but still substantial gains, indicating the broad applicability of our STRUCTURE across different alignment methods.

---

[3] $\lceil 2\sqrt[3]{N} \rceil$, which comes from the literature of choosing the optimal number of histogram bins [32].

Table 2: Performance comparison of zero-shot classification and cross-modal retrieval tasks across eight datasets and three alignment methods. Alignment is performed for a RoBERTa and a ViT-Giant. Underline shows the best performance in the respective group (*i.e.*, alignment method), **bold** indicates the overall best performance for the respective dataset, and *italic* indicates our framework.

| Method | Zero-shot Classification (Accuracy) | | | | | | | Retrieval (R@1) | |
|---|---|---|---|---|---|---|---|---|---|
| | STL10 | CIFAR10 | Caltech101 | Food101 | CIFAR100 | ImageNet | Pets | Flickr30 I2T | Flickr30 T2I |
| Linear + Last [10] | 75.6 | 85.5 | 37.9 | 14.8 | 34.0 | 9.9 | 7.0 | 32.5 | 22.1 |
| *Linear + Similar* | 79.7 | 89.0 | 39.5 | 14.6 | 33.1 | 10.5 | 4.9 | 35.3 | 24.0 |
| *Linear + Similar + $\mathcal{R}_S$* | 92.6 | 96.3 | 56.0 | **30.6** | 51.3 | 24.7 | 13.2 | 65.8 | 53.7 |
| MLP + Last [9] | 76.6 | 79.2 | 38.2 | 15.6 | 35.3 | 10.6 | 5.3 | 31.6 | 20.3 |
| *MLP + Similar* | 84.0 | 81.5 | 38.8 | 17.1 | 34.5 | 11.4 | 6.1 | 36.4 | 25.0 |
| *MLP + Similar + $\mathcal{R}_S$* | **92.7** | 96.3 | 56.0 | 30.5 | 52.1 | 25.1 | 13.2 | **65.9** | **53.8** |
| CSA + Last [29] | 77.9 | 78.5 | 31.4 | 29.3 | 47.4 | 23.2 | 14.4 | 47.0 | 38.3 |
| *CSA + Similar* | 80.0 | 80.8 | 33.6 | 28.0 | 47.4 | 23.3 | 14.9 | 48.6 | 39.0 |
| *CSA + Similar + $\mathcal{R}_S$* | 91.7 | **97.2** | **61.5** | 28.6 | **56.4** | **26.8** | **17.0** | 56.1 | 43.1 |

In addition to regularization, selecting layers based on representational similarity instead of defaulting to the final layer also leads to consistent improvements across all methods. On average, this strategy yields relative gains in classification accuracy of $2.5\%$ for Linear, $4.8\%$ for MLP, and $2.0\%$ for CSA. For retrieval tasks, the relative improvements are even more pronounced, $8.6\%$ for Linear, $18.3\%$ for MLP, and $2.7\%$ for CSA. This strategy alone provides a significant boost in performance and, when combined with STRUCTURE, results in the best overall generalization. The complete results across all 24 datasets are shown in Appendix H.8, showing consistent trends on the extended benchmark: average improvements from baseline to our approach are $65.0\%$ for Linear in classification and $122.7\%$ in retrieval, $61.5\%$ and $136.8\%$ for MLP, and $28.3\%$ and $15.9\%$ for CSA, respectively. Interestingly, on the CIFAR10 dataset, our alignment strategy even outperforms CLIP [7] by around $2\%$ while using only $0.02\%$ of the data used to train CLIP.

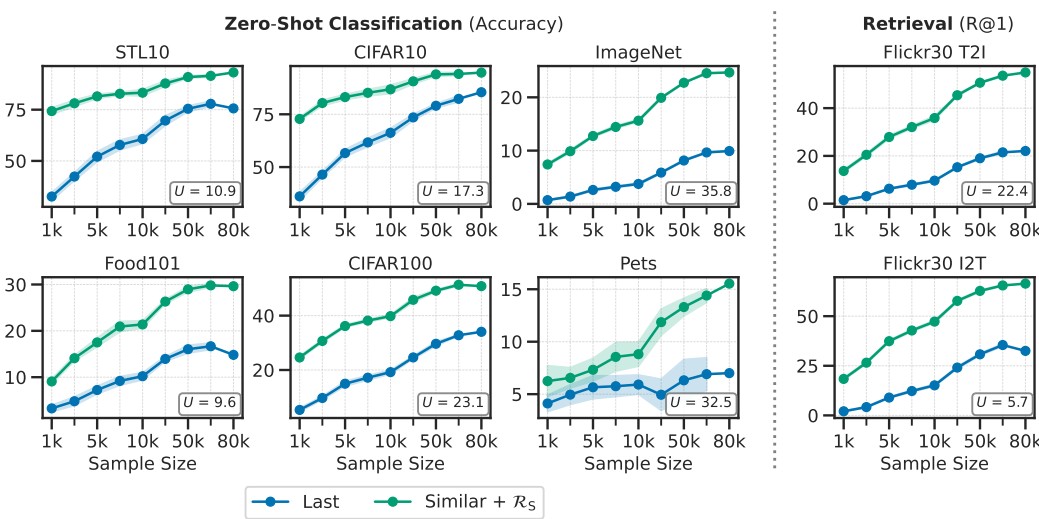

Figure 3: Comparison of zero-shot and retrieval performance for linear alignment when scaling down the training data, repeated five times for each sample size. Here, $U$ quantifies the proposed method's label efficiency by computing the utility compared to using the last layer.

**Scaling down the training data.** To measure the effect of the size of the training dataset on the performance, we subsampled the COCO training set and evaluated the effectiveness of the proposed method across varying dataset sizes for both zero-shot classification and retrieval tasks. We observe that across different tasks and sample sizes, STRUCTURE regularization combined with layer selection significantly improves performance, even in extreme resource-constrained settings of only 1,000 samples (see Figure 3). Similarly, we find that the layer selection strategy continues to provide improvements even with less data (Appendix H.3). These results demonstrate that combining

latent-geometry preservation with strategic layer selection is highly performant under severe data scarcity, making multimodal alignment practical even in the most constrained low-data regimes.

To quantify label efficiency, we measure the utility [35] $U(N) = (\hat{N}/N) - 1$ for different-sized subsets and report its average value. Here, $N$ is the number of labeled examples used by the regularized model, and $\hat{N}$ is the minimal number of labeled examples required when naively relying on the last layer for alignment without any regularization to match that same accuracy. This metric tracks the reduction in samples needed to achieve equivalent performance. For example, for zero-shot classification on the CIFAR100 dataset and text-to-image retrieval performance on the Flickr30 dataset, the outlined method yields an average utility of $23.1\times$ and $22.4\times$, respectively, indicating that roughly $23\times$ fewer samples are needed to achieve the same performance as the baseline. These substantial label savings demonstrate that our approach significantly reduces annotation requirements while preserving high performance, thereby enhancing practicality for real-world multimodal applications.

**Scaling up the training data.** We next explored the effect of increasing the size of the multimodal dataset used for alignment. We gradually incorporate increasingly larger subsets of the LAION-15M dataset [36] into the COCO dataset of $80K$ samples. Following the approach of Gadre et al. [37], we filter out low-quality LAION samples by computing CLIP scores [38] for each image–text pair and discarding those with a score below $0.15$. From the remaining $847K$ samples, we randomly draw subsets and incrementally add them to the $80K$ COCO training set in steps of 80K. Figure 4 presents the results of incrementally

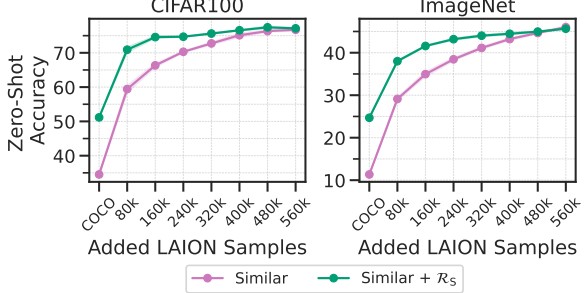

Figure 4: Zero-shot performance when randomly adding LAION samples to the COCO training set, repeated three times, when aligning the best layers and adding the regularization.

adding different numbers of paired samples for CIFAR100 and ImageNet zero-shot classification, and we compare performance with and without applying STRUCTURE regularization on layers selected based on representational similarity. The steepest performance gain occurs with the initial addition of 80K samples, while further increases yield smaller but steady improvements. Notably, the impact of STRUCTURE regularization diminishes as more data becomes available, highlighting its particular effectiveness in low-data regimes, where preserving pretrained structure is most beneficial.

**Train-test data distribution shift.** Despite the advantages of our alignment approach in low-data regimes, performance remains low on certain datasets. For example, accuracy on the Pets dataset reaches only around $15\%$. We attribute this to distribution shifts between the COCO-based training set and the target evaluation domains, including differences in image content, label granularity, and overall dataset coverage and scale. To understand the impact of this shift, we augmented the training set (80K COCO pairs) with a small number of $k$ in-domain examples per class drawn from different image classification benchmarks. These $k$ samples per class are added directly to the 80,000-sample COCO training set (e.g., $80,000 + 10 \times k$ total samples for CIFAR10). We find that mixing a small number of samples from the target domain into the training set can effectively close distribution gaps and achieve strong performance. Figure 5 plots zero-shot accuracy as a function of number of samples per class $k$ across five datasets, including Flowers, Food101, CIFAR100, Pets, and ImageNet. Remarkably, we can achieve or even outperform CLIP trained on hundreds of millions of multimodal samples by adding a few in-distribution labeled samples. Specifically, with just three samples per class, accu-

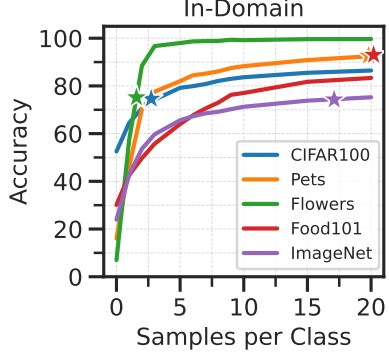

Figure 5: Zero-shot classification performance as in-domain samples are added to the training set. Performance is evaluated on multiple fine-grained in-domain tasks, where the added and evaluation samples come from the same dataset. Here, the star indicates the CLIP performance.

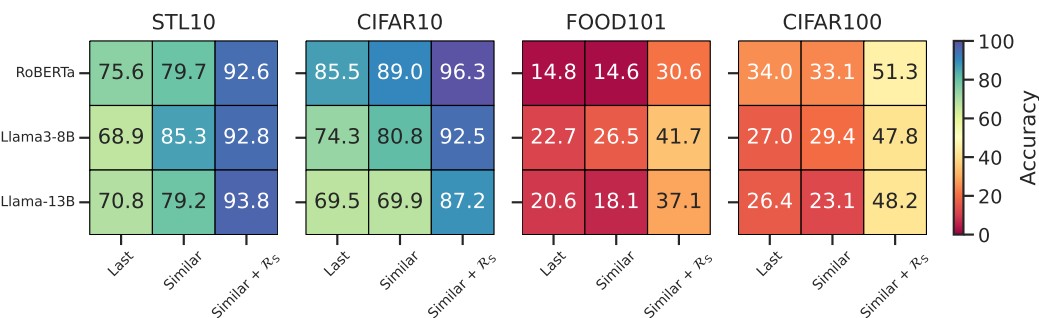

Figure 7: Performance of different language models combined with a DINOv2 ViT-G when choosing the last or most similar layer for alignment, and the influence of adding STRUCTURE regularization when training a single linear alignment layer.

racy on the Flowers dataset rises from around $24\%$ to over $95\%$, outperforming CLIP's $93\%$ accuracy. Similarly, on the CIFAR100 dataset, using only four samples per class yields a $3\%$ improvement over CLIP. On the ImageNet dataset, our alignment strategy matches CLIP's performance with only 17 samples per class. On both Pets and Flowers datasets, we achieve comparable results to CLIP using approximately 20 samples per class. These results suggest that in a limited data regime, labeling just a few in-domain samples can yield comparable or even superior performance to large models trained on millions of paired samples.

**Different pairs of models.** We explore the performance across different language models, each combined with a DINOv2 ViT-G encoder. Figure 7 shows that our regularization term and layer selection strategy consistently improve results across all combinations of models. On average, our method yields improvements of $15.2\%$ for RoBERTa, $20.5\%$ for Llama3-8B, and $19.8\%$ for Llama-13B. Notably, the RoBERTa-based combination achieves the highest overall performance, supporting previous findings [9] that RoBERTa aligns particularly well with vision FMs.

**Neighborhood preservation.** To empirically verify that the STRUCTURE regularization preserves pretrained neighborhood, we monitored Trustworthiness$_{100}$ and Continuity$_{100}$ [39] on a fixed random subset of $5,000$ embeddings from the training and the held-out validation set at the end of each epoch with and without the STRUCTURE regularization. Trustworthiness measures the fraction of $k$-nearest neighbors in the aligned space that were already neighbors in the original pretrained embedding, while Continuity measures the fraction of original neighbors that remain in the aligned space. As shown in Figure 6, over 2,000 epochs, both the training Trustworthiness and Continuity steadily decline under the standard alignment objective, while the gap between training and validation values steadily widens. In contrast, with STRUCTURE regularization, Trustworthiness and Continuity curves on training and validation remain between $0.99$

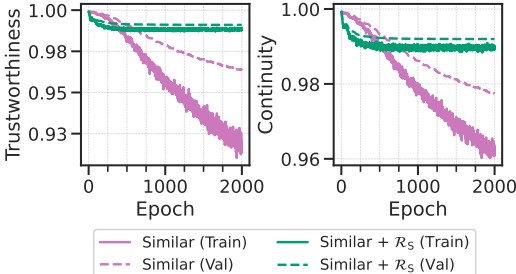

Figure 6: Evolution of Trustworthiness$_{100}$ and Continuity$_{100}$ over $2,000$ training epochs on a fixed random subset of $5,000$ embeddings from the training (solid lines) and validation set (dashed lines) when aligning the most similar layers without and with regularization. Trustworthiness penalizes introducing new neighbors not present in the original pretrained space, while continuity penalizes losing true (pretrained) neighbors.

and $1.00$ (gap $< 0.002$) with no drift. This demonstrates that our regularizer enforces consistent, geometry-respecting alignments throughout optimization, avoiding the over-warping seen in unregularized alignment.

**Further experiments.** Additional results are provided in the Appendix H, including experiments with text–audio alignment and application to the biological domain (see H.4), more ablations such as the robustness to the parameter $\lambda$, number of regularization levels $L$, layer selection metric,

normalization schemes, and distance functions (see H.6), as well as comparisons to unsupervised alignment methods (see H.7).

## 6    Conclusion

We present a simple, yet effective, framework for parameter-frozen modality alignment in low-data regimes, leveraging two key components: *(i)* a novel STRUCTURE regularizer that preserves the multi-scale neighborhood geometry of each modality's pretrained latent space, and *(ii)* an automatic layer-selection procedure that identifies and aligns the pair of intermediate layers with the highest representation similarity. Both components integrate seamlessly with existing alignment pipelines, be they linear projections, MLPs, or advanced matrix-decomposition methods, offering a plug-and-play solution for any modality alignment method. We evaluate our strategy by incorporating it into three existing modality alignment methods and observe consistent performance improvements across 24 benchmark datasets covering zero-shot classification and retrieval tasks, with average relative improvements of $51.6\%$ in classification and $91.8\%$ in retrieval tasks. This stems from a faithful preservation of pretrained geometry throughout training, enforced by the proposed regularization. We further show that injecting only a handful of in-domain examples per class achieves comparable performance to multimodal models trained on hundreds of millions of samples.

## Limitations

While our method performs competitively in the low-data regime, there remains a performance gap on more challenging tasks compared to models like CLIP that are trained on hundreds of millions of paired samples. That said, we show that this gap can be significantly reduced by incorporating only a few in-domain samples in Figure 5, which highlights the potential of few-shot adaptation. Additionally, this prompted us to consider which data is most effective for aligning the spaces of different modalities. A promising future direction is to systematically investigate how different types of data influence cross-modal alignment quality in a low-data regime.

Currently, we have only investigated aligning two modalities. However, our framework can be straightforwardly extended to three or more modalities. Since the STRUCTURE regularization is computed independently for each modality to preserve its own pretrained structure, this extension only requires adding an additional regularization term to the loss function for each new modality. For instance, aligning three modalities $(X_1, X_2, X_3)$ would involve the following objective:

$$\mathcal{L} = \mathcal{L}_A + \lambda(\mathcal{R}_S^{(L)}(X_1, f_1(X_1)) + \mathcal{R}_S^{(L)}(X_2, f_2(X_2)) + \mathcal{R}_S^{(L)}(X_3, f_3(X_3))).$$

Since aligning more than two modalities is still a very active research topic, we leave these investigations to future efforts.

## Acknowledgements

We thank Artyom Gadetsky, Ramon Vinas Torné, Myeongho Jeon, Luca Viano, and Simone Lionetti for their valuable suggestions, which helped improve the clarity of the manuscript. We gratefully acknowledge the support of the Swiss National Science Foundation (SNSF) starting grant TMSGI2_226252/1, SNSF grant IC00I0_231922, and the CIFAR MHU Catalyst.

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

## A    Computational Resources

Our experiments used a cluster of 8 NVIDIA GeForce RTX 3090 GPUs, but each individual training run required only a single GPU and less than 4GB of VRAM and took at most 2 hours. The most computationally demanding part was obtaining the embeddings from the pretrained unimodal encoders, which on a single GPU with a batch size of 16 has a speed of 3 sec/iter.

## B    Implementation

```python
import torch
import torch.nn.functional as F

def reg_structure(X, A, L=1, tau=.05, eps=1e-8):
    X_hat = F.normalize(X, p=2, dim=1, eps=eps)
    A_hat = F.normalize(A, p=2, dim=1, eps=eps)

    X_tilde = X_hat - X_hat.mean(dim=0, keepdim=True)
    A_tilde = A_hat - A_hat.mean(dim=0, keepdim=True)

    Sx = (X_tilde @ X_tilde.T) / tau
    Sa = (A_tilde @ A_tilde.T) / tau

    Px = F.softmax(Sx, dim=1)
    Pa = F.softmax(Sa, dim=1)

    r_S = 0.0
    for l in range(1, L + 1):
        Px_l, Pa_l = Px.matrix_power(l), Pa.matrix_power(l)
        M_l = 0.5 * (Px_l + Pa_l)
        d_js = 0.5 * ((Pa_l * (torch.log(Pa_l + eps) - torch.log(M_l + eps))).sum()
                    + (Px_l * (torch.log(Px_l + eps) - torch.log(M_l + eps))).sum())
        r_S += d_js / l
    return r_S / L
```

Listing 1: PyTorch reference implementation of the STRUCTURE regularizer $\mathcal{R}_\mathrm{S}(X, A)$

## C    Proof of the generalization gap of STRUCTURE regularization

Unlike classical weight-space penalties such as the $\ell_1$ norm, which impose sparsity directly on the model parameters and are agnostic to the number of training examples available, the proposed STRUCTURE regularizer is constructed in the *sample* space, via pairwise similarities among the $N$ input embeddings. Consequently, its numerical value and statistical behavior depend explicitly on $N$, the size of the dataset or batch size, respectively. To ensure that this penalty remains well-behaved as the number of samples changes, we introduce its *population counterpart*, defined as the expectation of the empirical similarity-based regularizer under the true data distribution. In the following section, we will show that the empirical estimator is unbiased and concentrates around its population expectation at a rate of $\mathcal{O}(1/\sqrt{N})$, thereby providing a computationally efficient yet statistically reliable approximation to the ideal regularization term.

For a fixed number of hierarchical levels $L$, we define the *STRUCTURE regularizer*

$$\mathcal{R}_\mathrm{S}^{(L)}(X, A) = \frac{1}{L} \sum_{l=1}^{L} \frac{\mathrm{JS}(P_X^{(l)}, P_A^{(l)})}{l}, \tag{10}$$

where $P_X = \mathrm{softmax}(S_X) = \mathrm{softmax}\left(\frac{\tilde{X}\tilde{X}^\top}{\tau}\right)$, $P_A$ is defined analogously, and $P_X^{(l)} = (P_X)^l$, $P_A^{(l)} = (P_A)^l$.

We define the empirical and expected values as

$$\hat{\mathcal{R}}_N = \mathcal{R}_\mathrm{S}(X, A), \quad \mathcal{R}^\star = \mathbb{E}_{(x_1, x_2) \sim \mathcal{D}}\left[\hat{\mathcal{R}}_N\right].$$

In order to prove the generalization gap of the regularization, we investigate how much each sample pair can influence the output and use this to derive the bound using McDiarmid's inequality.

**Lemma 2** (Per-sample sensitivity). *Replacing a single pair $(x_1^i, x_2^i)$ by an arbitrary $(\tilde{x}_1^i, \tilde{x}_2^i)$, while keeping the other $N-1$ pairs fixed, changes the value of* (10) *by at most*

$$\Delta_N = \frac{4 \log 2}{N}. \tag{11}$$

*Proof.* Replacing a single sample $(x_1^i, x_2^i)$ alters exactly one row and one column of each similarity matrix $S_X, S_A$, which after row-wise softmax means exactly two rows of $P_X$ (and two of $P_A$) can change. We measure the matrix-level deviation by averaging the total variation (TV) distance of corresponding rows,

$$d(P, P') = \frac{1}{N} \sum_{j=1}^{N} \mathrm{TV}\left(P(j, \cdot), P'(j, \cdot)\right), \quad \mathrm{TV}(p, p') = \tfrac{1}{2} \sum_i |p_i - p_i'| \leq 1,$$

so changing two rows gives $d(P_X, P_X') \leq \frac{2}{N}$ and similarly $d(P_A, P_A') \leq \frac{2}{N}$. Next, we check the influence of the sample perturbation for the Jensen–Shannon divergence

$$\mathrm{JS}(P_X, P_A) = \tfrac{1}{2} D_{\mathrm{KL}}\left(P_X \| M\right) + \tfrac{1}{2} D_{\mathrm{KL}}\left(P_A \| M\right),$$

where $M = \frac{1}{2}(P_X + P_A)$. Each KL term is bounded by $\log 2$, and more generally if $\mathrm{TV}(p, p') \leq \delta$ and $\mathrm{TV}(q, q') \leq \delta$ then $\left|\mathrm{JS}(p, q) - \mathrm{JS}(p', q')\right| \leq \log 2 [\mathrm{TV}(p, p') + \mathrm{TV}(q, q')]$. Hence, at any fixed level $L$, the two-matrix, two-row perturbation shifts the JS divergence by at most $\log 2(\frac{2}{N} + \frac{2}{N}) = \frac{4 \log 2}{N}$, and averaging over $l = 1, \ldots, L$ with weights $1/l$ cannot increase this bound. $\square$

**Lemma 3** (Generalization bound). *The generalization gap between the empirical and the expected values of $\mathcal{R}_S$ is bounded by*

$$\left|\hat{\mathcal{R}}_N - \mathcal{R}^\star\right| \leq \mathcal{O}\left(\frac{1}{\sqrt{N}}\right). \tag{12}$$

*Proof.* McDiarmid's bounded-difference inequality states that for any function $f$ of independent variables $X_1, X_2, \ldots, X_N$,

$$\Pr\left(\left|f(X_1, X_2, \ldots, X_N) - \mathbb{E}[f(X_1, X_2, \ldots, X_N)]\right| \geq \varepsilon\right) \leq 2 \exp\left(-\frac{2\varepsilon^2}{\sum_{i=1}^{N} c_i^2}\right),$$

where $c_i$ bounds the change in $f$ when only the $i^{\text{th}}$ input is perturbed. Using $c_i = \Delta_N$ from (11),

$$\sum_{i=1}^{N} c_i^2 = N \Delta_N^2 = N \left(\frac{4 \log 2}{N}\right)^2 = \frac{16 \log^2 2}{N}.$$

Hence, for every $\varepsilon > 0$,

$$\Pr\left(\left|\hat{\mathcal{R}}_N - \mathcal{R}^\star\right| \geq \varepsilon\right) \leq 2 \exp\left(-\frac{\varepsilon^2 N}{8 \log^2 2}\right). \tag{13}$$

Next, we fix a confidence level $1 - \delta$ with $0 < \delta < 1$ and set the right-hand side of (13) to $\delta$. Solving for $\varepsilon$ gives

$$\varepsilon = 2\sqrt{2} \log 2 \sqrt{\frac{\log(2/\delta)}{N}}.$$

Thus, with probability at least $1 - \delta$,

$$\left|\hat{\mathcal{R}}_N - \mathcal{R}^\star\right| \leq 2\sqrt{2} \log 2 \sqrt{\frac{\log(2/\delta)}{N}} \tag{14}$$

showing $\mathcal{O}\left(\frac{1}{\sqrt{N}}\right)$ convergence. $\square$

# D  Hyperparameters

Table 3 lists all default hyperparameters that were used throughout the paper if not otherwise specified.

Table 3: List of default hyperparameters used throughout the paper.

| Category | Hyperparameter | Value |
|---|---|---|
| Layer selection | Validation size | 5,000 |
| | Metric | Mutual kNN (k=rice) |
| Alignment training | Epochs | 1,000 |
| | Batch size | 4,096 |
| | Learning rate scheduler | Cosine |
| | Auto learning rate finder | [40] |
| | Gradient clipping | 1.0 |
| | Early stopping epochs | 200 |
| | Optimizer | AdamW |
| | Weight decay | 0.0001 |
| Alignment objective | Temperature $\tau$ | 0.05 |
| | $\mathcal{R}_S$ levels $L$ | 1 |
| | $\lambda$ | 10.0 |
| | $\lambda$ warmup | Linear |
| | $\lambda$ warmup steps | 1,000 |
| Alignment layer | Output dimension | 512 |

# E    Description of evaluation datasets

We use 22 vision datasets for evaluation of zero-shot classification and two retrieval datasets for evaluation on retrieval tasks, similar to those studied in Radford et al. [7], except for four tasks because of either un-availability of the dataset or lack of diversity in the modality-specific pretrained spaces, leading to random performance. These datasets cover a wide range of vision tasks, including general object classification datasets CIFAR10 [41], CIFAR100 [41], STL10 [42], ImageNet [43], Caltech101 [44]; fine-grained object classification datasets Food101 [45], Flowers [46], Cars [47], FGVC Aircraft [48], Pets [49]; handwritten digits classification dataset MNIST [50]; texture classification dataset DTD [51]; scene classification dataset SUN397 [52]; the facial emotion recognition dataset FER2013 [53]; the satellite image classification datasets EuroSAT [54], Resisc45 [55]; the German Traffic Sign Recognition Benchmark (GTSRB) [56]; the KITTI Distance dataset [57]; the metastatic tissue classification dataset PatchCamelyon (PCam) [58]; action recognition datasets UCF101 [59], Kinetics700 [60]; the country classification dataset Country211 [7]. Furthermore, we include two standard image–text retrieval benchmarks: MS COCO [33] and Flickr30 [61]. For the two video datasets, UCF101 and Kinetics700, we take the middle frame of each video clip as the input of the pre-trained models. We use *accuracy* for zero-shot classification evaluation and *recall@1* (R@1) for retrieval evaluation. For zero-shot classification evaluation, we follow the same setup and use the same prompts as in Radford et al. [7].

# F    Description of pretrained unimodal encoders

We compare performance when using three self-supervised vision encoders from the DINOv2 family for alignment. Namely, a Vision Transformer Base (ViT-B), Large (ViT-L), and Giant (ViT-G), where each is pretrained on massive unlabeled image corpora using distilled masked prediction objectives to produce rich, high-dimensional patch-level embeddings [3]. For language, we compare RoBERTa, a transformer-based encoder optimized on large-scale English text, which outputs contextualized token representations that are aggregated into a fixed-length sentence embedding [34]. In addition, we leverage two sizes of the Llama3 decoder-only transformer family, namely an 8B and 13B parameter version, which are pretrained on web-scale text data for generative tasks [2]. In all experiments, we freeze the unimodal encoders and learn only lightweight projection layers atop their latent outputs to align them into a shared multimodal embedding space.

# G    Layers with the highest similarity

Table 5 lists the model combinations in this work along with the layers of each encoder that have the highest representational similarity measured in terms of mutual kNN with k chosen according to Rice's criterion.

Table 4: List of benchmarks we used for both zero-shot classification and retrieval evaluation.

| Task | Dataset | Number of Classes | Train size | Test size |
|---|---|---|---|---|
| Classification | Food101 [45] | 101 | 75,750 | 25,250 |
| | CIFAR10 [41] | 10 | 50,000 | 10,000 |
| | CIFAR100 [41] | 100 | 50,000 | 10,000 |
| | SUN397 [52] | 397 | 19,850 | 19,850 |
| | Cars [47] | 196 | 8,144 | 8,041 |
| | FGVC Aircraft [48] | 100 | 6,667 | 3,333 |
| | DTD [51] | 47 | 3,760 | 1,880 |
| | OxfordPets [49] | 37 | 3,680 | 3,669 |
| | Caltech101 [44] | 102 | 3,060 | 6,084 |
| | Flowers [46] | 102 | 2,040 | 6,149 |
| | MNIST [50] | 10 | 60,000 | 10,000 |
| | FER2013 [53] | 7 | 28,709 | 3,589 |
| | STL10 [42] | 10 | 5,000 | 8,000 |
| | EuroSAT [54] | 10 | 10,000 | 5,000 |
| | Resisc45 [55] | 45 | 25,200 | 6,300 |
| | GTSRB [56] | 43 | 26,640 | 12,630 |
| | KITTI Distance [57] | 4 | 5,985 | 1,496 |
| | Country211 [7] | 211 | 42,200 | 21,100 |
| | PatchCamelyon [58] | 2 | 294,912 | 32,768 |
| | UCF101 [59] | 101 | 9,537 | 3,783 |
| | Kinetics700 [60] | 700 | 536,485 | 33,966 |
| | ImageNet [43] | 1000 | 1,281,167 | 50,000 |
| Retrieval | MS COCO [33] | N/A | 82,783 | 40,504 |
| | Flickr30 [61] | N/A | 29,783 | 5,000 |

Table 5: Layer combination with the highest representational similarity in terms of mutual kNN, where k is chosen according to Rice's criterion.

| Language Model | Vision Model | Language Index | Vision Index |
|---|---|---|---|
| RoBERTa | ViT-L | 24 | 22 |
| | ViT-G | 24 | 38 |
| Llama3-8B | ViT-G | 25 | 36 |
| Llama13B | ViT-G | 29 | 36 |

# H   Further results

In this section, we provide a detailed results of the experiments presented in the main paper. In Section H.7, we compare our supervised alignment method to an unsupervised approach, Section H.6 presents details on the ablation study that isolates the effects of regularization and layer selection, Section H.2 analyzes the consistency of our layer selection procedure across different validation set sizes, and Section H.8 presents detailed results of the downstream performance on zero-shot classification and retrieval tasks.

## H.1   Layer-wise performance

In Section 4, we propose selecting layers for alignment based on their representational similarity rather than defaulting to the final layers. To provide a more comprehensive view, Figure 8 shows a detailed layer-wise analysis between the last five layers of a RoBERTa encoder and a DINOv2 ViT-L encoder in terms of the respective downstream performance these combinations achieve after alignment. Results show that the most suitable layers (here 24/22 for the language and vision encoder pair) yield strong performance across all datasets. Additionally, we can see that for CIFAR10 and UCF101, there is a tendency that later vision layers lead to better performance while the choice of the language layers is less influential.

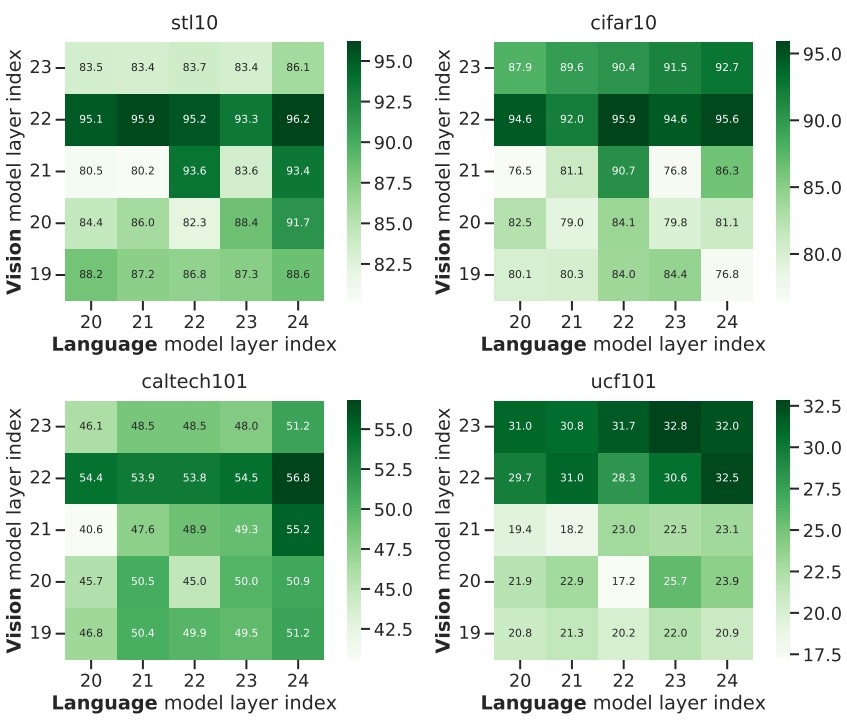

Figure 8: Downstream performance of the last five layers for a RoBERTa encoder and a DINOv2 ViT-L encoder.

## H.2 Layer selection consistency

Figure 9 evaluates the stability of layer selection by repeating the process 100 times on random validation subsets of varying sizes ($N = 100, 500, 1000,$ and $5000$) for mutual kNN with k=rice. All metrics illustrate high robustness against repeated selection on different subsets, justifying our choice of a 5000-sample set to ensure reliable layer pairing across runs.

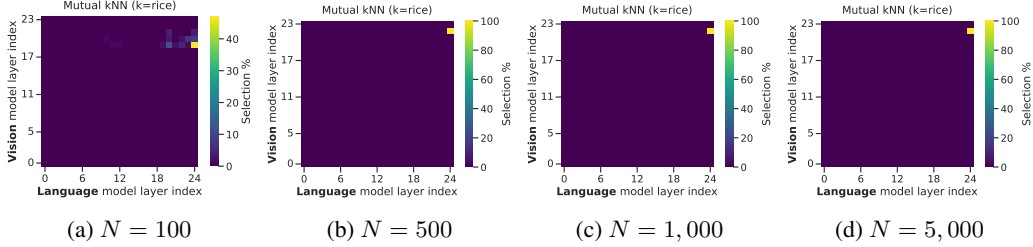

Figure 9: Consistency of different mutual kNN (k=rice) when repeatedly selecting $N$ samples and selecting the best layer combination.

## H.3 Low-data comparison layer selection

Figure 10 compares the layer selection strategy when scaling down the training data using the same setup as for Figure 3 of the main paper. Results show a minor difference in performance and utility when using the most similar layers for alignment. However, results show that the proposed layer selection strategy is a simple addition to existing alignment methods as the method includes choosing the last layers to align if they indeed are the most similar. This can be clearly seen as the performance across tasks is positive. Thus the selection process can additionally boost performance at minimal additional overhead.

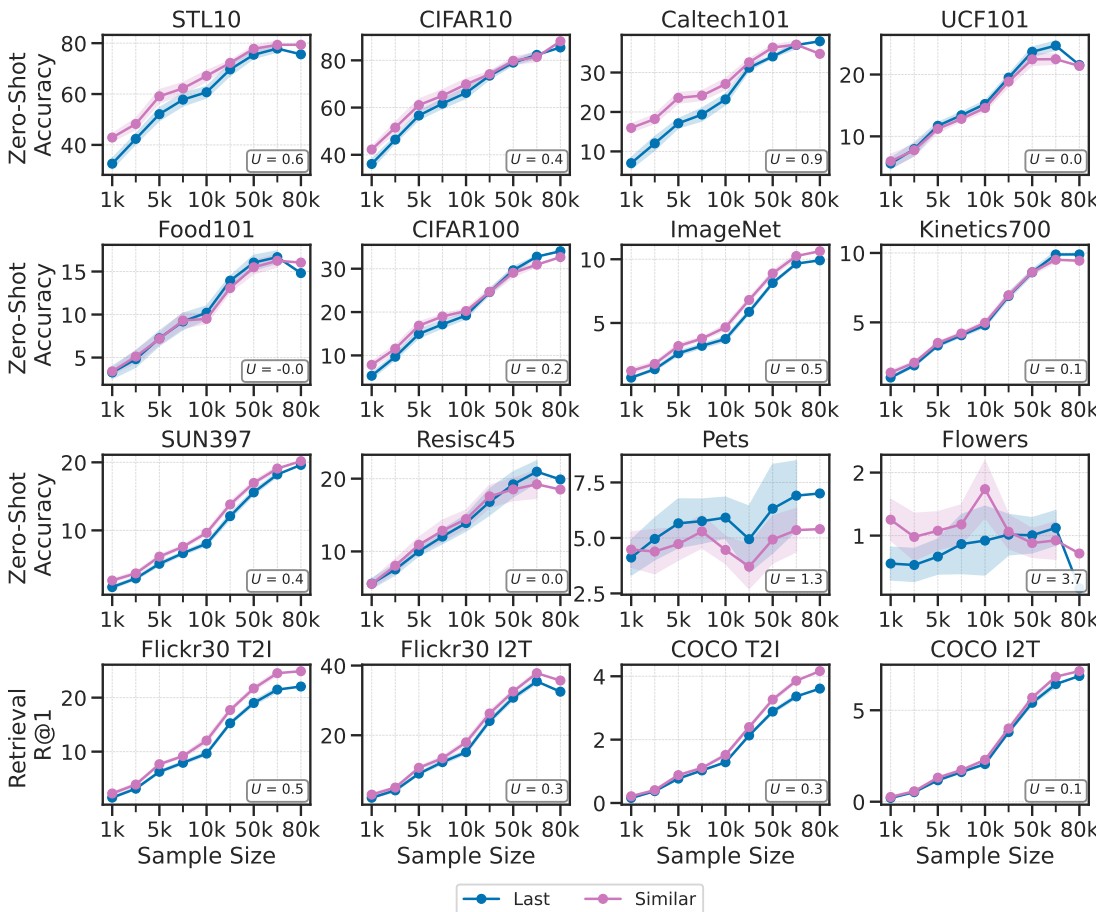

Figure 10: Zero-shot and retrieval performance for training linear alignment using the last or most similar layer. Here, $U$ quantifies the proposed method's label efficiency by computing the utility compared to using the last layer.

## H.4 Application to other modalities

**Audio-text.** We further evaluate the proposed method for audio-language alignment. Here, as a pretrained unimodal encoder, we use BEATs-iter3 [62] for audios, and extract text embeddings using RoBERTa [34]. We train a linear alignment on 3,839 audio-caption pairs from the Clotho dataset [63].

To evaluate cross-modal alignment, we perform audio-to-text and text-to-audio retrieval on a held-out set of 1,045 pairs from the same dataset. As shown in Table 6, our STRUCTURE regularization significantly improves the retrieval performance compared to the baseline without regularization. This demonstrates the generalizability of our regularization method across modalities beyond vision–language alignment.

Table 6: Retrieval performance (R@k) for audio–text alignment.

| Method | Audio-to-Text | | Text-to-Audio | |
|---|---|---|---|---|
| | R@1 | R@5 | R@1 | R@5 |
| Linear + Last [10] | 0.2 | 1.2 | 0.4 | 2.9 |
| Linear + Similar | 0.6 | 2.0 | 0.0 | 1.4 |
| Linear + Similar + $\mathcal{R}_S$ | **7.9** | **23.6** | **8.0** | **24.6** |

**Single-cell image-transcriptome.** In addition to the audio application, we have conducted additional experiments on the biological domain using only 19,900 paired human single-cell image-transcriptome data. In

this experiment, we linearly align a UCE [64] encoder for the transcriptome data and a masked autoencoder (MAE) trained on single-cell images for the image data. We split the 19,900 paired samples into 100 training samples and 19,800 test samples to simulate the low data setting. Results below show that our regularization also helps in the biological domain, yielding relative improvements of 43.1% on the retrieval task and 30.3% on the cell type classification task, even if the Platonic representation hypothesis is not guaranteed in the biological domain.

Table 7: Retrieval and classification performance for single-cell image-transcriptome alignment

| Method | Retrieval (R@5) | Classification |
|---|---|---|
| Linear + Last [10] | 4.2 | 40.3 |
| Linear + Similar + $\mathcal{R}_S$ | **6.0** | **52.5** |

## H.5 Modality Gap Analysis

We measured the modality gap [65] for three different alignment setups for a ViT-Giant and RoBERTa Large, trained on COCO with a linear alignment function. The modality gap was measured for the two image-text datasets considered in our submission (i.e., COCO and Flickr30) to ensure that the comparison is influenced solely by the data itself and not by any other artifacts introduced by the evaluation, such as a zero-shot template, as would be necessary for the classification datasets. Interestingly, the results given in Table 8 show that the modality gap indeed decreases when first selecting the most similar layers and then further when applying STRUCTURE regularization. This aligns with the findings from the respective paper [65], that the modality gap indeed has a relationship with downstream performance.

Table 8: Modality gap for different alignment setups on the COCO and Flickr30 test sets. The gap was measured as the Euclidean distance between the mean of the image embeddings and the mean of the text embeddings after $L_2$ normalization.

| Method | COCO | Flickr30 |
|---|---|---|
| Linear + Last [10] | 6.6 | 6.8 |
| Linear + Similar | 6.0 | 6.3 |
| Linear + Similar + $\mathcal{R}_S$ | 4.5 | 3.8 |

## H.6 Ablation study

Table 9 examines the influence of different hyperparameters of the proposed method. Specifically the number of hierarchical levels in the STRUCTURE regularizer, regularization strength $\lambda$, layer selection strategy, normalization schemes, and distance functions. Varying the number of levels from one to five yields virtually identical accuracy across datasets, indicating that a single-level penalty suffices to preserve pretrained geometry. A strong regularization weight $\lambda = 10$ gives the best performance (*e.g.*, STL10 at 92.6%, CIFAR10 at 96.3%), whereas $\lambda \in \{0.1, 1\}$ under-regularizes leading to worse results. Without regularization, choosing the last layer yields 75.6% on STL10 and 85.5% on CIFAR10, but using mutual kNN layer selection boosts these to 79.7% and 89.0%, respectively. These results confirm that both strategic layer choice and sufficient regularization are critical for maximally leveraging low-data regimes. Additionally, Table 12 and Figure 7 from the main paper confirm that best-layer alignment and STRUCTURE regularization consistently increase performance across four language-vision model pairs. For normalization schemes and distance functions, STRUCTURE demonstrates robustness to these changes, highlighting that the primary benefit stems from the idea of regularizing hierarchical relationships.

The variable $N$ in the default setting of STRUCTURE represents the size of the batch, typically set to $4,096$ random samples (here approximately 6% of the entire dataset), which allows for a sufficient estimate of the true data generation distribution while maintaining computational efficiency. In Table 9, we also conducted experiments to demonstrate how the choice of sample size affects performance. Results show that the benefit of the regularization increases with increased sample size, but also show its efficiency, since even with 128 samples, there is a clear improvement. Thus a practical choice is to use bigger batch sizes if possible. Additionally we investigated the choice of randomly selecting the subset of samples used to compute the regularization. Results show that the regularization yields very similar results regardless of whether it uses a fixed (same-sized) or a random subset of the data.

Table 9: Ablation of components for two coarse-grained (STL10, CIFAR10) and two fine-grained tasks (Food101, CIFAR100), when applying STRUCTURE on different layers $l$, changing the contribution of $\mathcal{R}_S$, choosing the optimal layers according different metrics, changing the size and the type of subset used to compute the regularization, changing normalization, and using different distance functions. All experiments are performed for linear alignment.

| Ablation | STL10 | CIFAR10 | Food101 | CIFAR100 |
|---|---|---|---|---|
| *Regularization levels* | | | | |
| $\mathcal{R}_S^{(1)}$ | **92.6** | 96.3 | **30.6** | **51.3** |
| $\mathcal{R}_S^{(2)}$ | 92.5 | **96.4** | **30.6** | **51.3** |
| $\mathcal{R}_S^{(3)}$ | **92.6** | 96.3 | **30.6** | 51.1 |
| $\mathcal{R}_S^{(5)}$ | 92.5 | 96.1 | **30.6** | 51.0 |
| *Regularization strength* | | | | |
| $\lambda = 0.1$ | 90.8 | 95.1 | **30.9** | 50.2 |
| $\lambda = 1.0$ | 91.4 | 95.9 | 30.8 | 50.8 |
| $\lambda = 10.0$ | **92.6** | **96.3** | 30.6 | **51.3** |
| *Layer selection (w/o $\mathcal{R}_S$)* | | | | |
| Last | 75.6 | 85.5 | 14.8 | 34.0 |
| CKA [30] | **87.9** | 74.5 | 11.5 | 23.3 |
| Unbiased CKA [31] | **87.9** | 74.5 | 11.5 | 23.3 |
| Mutual kNN (k=10) [24] | 79.4 | 88.1 | **16.0** | 32.7 |
| Mutual kNN (k=rice) | 79.7 | **89.0** | 14.6 | **33.1** |
| *Batch size* | | | | |
| $N = 128$ | 90.3 | 96.5 | 29.2 | 48.6 |
| $N = 1,024$ | 92.1 | **96.8** | 30.2 | 50.4 |
| $N = 4,096$ | **92.6** | 96.3 | 30.6 | 51.3 |
| $N = 8,192$ | 92.5 | 96.5 | **31.0** | **52.1** |
| *Subset type* | | | | |
| random | **92.6** | 96.3 | 30.6 | 51.3 |
| fixed | **92.6** | **96.5** | **30.7** | **51.5** |
| *Normalization schemes* | | | | |
| Normalize + centering | 92.6 | 96.3 | 30.6 | 51.3 |
| Centering + normalize | 92.1 | 96.4 | 30.8 | 51.1 |
| Normalize | **92.7** | 96.1 | 30.5 | **51.5** |
| Standard scaling | 90.8 | **95.1** | **31.0** | 50.3 |
| *Distance functions* | | | | |
| Cosine | **92.6** | **96.3** | **30.6** | **51.3** |
| RBF | 90.8 | 94.9 | 30.9 | 50.3 |
| Spearman | 90.4 | 91.6 | 30.0 | 50.0 |

## H.7 Comparison to unsupervised methods

Table 10 compares our supervised alignment method (Linear + Similar + $\mathcal{R}_S$) with unsupervised Kernel Local CKA [12] and ASIF [23]. Across all three evaluation tasks (COCO, CIFAR-10, and CIFAR-100), our method yields significant improvement in both top-1 and top-5 accuracy while relying on only 80,000 paired samples. For instance, for COCO retrieval, Kernel Local CKA attains only 13.1% R@1, whereas the proposed method reaches 32.7% (a 19.6 pp. improvement) and improves R@5 from 40.0% to 56.2%. Similarly, on CIFAR10 our approach jumps from 0.06% to 96.8% top-1, and on CIFAR-100 from 1.13% to 46.9%. Compared against ASIF, results demonstrate that our method again substantially outperforms it, achieving relative improvements of 32.9% on the COCO retrieval task, 32.9% on the CIFAR-100 fine-grained classification task, and 5.6% on the CIFAR-10 coarse-grained classification task.

Table 10: Top-1 and Top-5 accuracy of unsupervised approaches compared against our framework on different datasets.

| Method | COCO | | CIFAR10 | | CIFAR100 | |
|---|---|---|---|---|---|---|
| | R@1 | R@5 | Top-1 | Top-5 | Top-1 | Top-5 |
| Kernel Local CKA [12] | 13.1 | 40.0 | 0.06 | 3.2 | 1.13 | 4.0 |
| ASIF [23] | 24.6 | 45.4 | 91.7 | 94.5 | 23.8 | 38.2 |
| Linear + Similar + $\mathcal{R}_S$ | **32.7** | **56.2** | **96.8** | **96.8** | **46.9** | **46.9** |

## H.8 Extended downstream results

Table 11 expands on the results in Table 2 from the main paper, detailing results on all evaluation tasks, including 22 zero-shot classification and two retrieval datasets with an additional fourth alignment technique (FuseMix [22]). Figure 11 expands on Figure 3 with more zero-shot classification and retrieval tasks. Figure 12 illustrates the performance of model combinations across different vision models, analogous to Figure 7 from the main paper, but with more combinations. Table 12 expands on Figure 7 from the main paper, detailing results on all evaluation tasks, including 22 zero-shot classification and two retrieval datasets.

Table 11: Performance for multiple evaluation tasks and alignment methods. Underline shows the best performance in the respective group (*i.e.*, alignment method), and **bold** indicates the overall best performance for the respective dataset. This table expands on table 2 from the main paper, detailing results on 22 zero-shot classification and two retrieval datasets.

| Method | STL10 | Caltech101 | Food101 | CIFAR10 | CIFAR100 | ImageNet | UCF101 | Kinetics700 | SUN397 | EuroSAT | Resisc45 | Flickr30 I2T | Flickr30 T2I |
|---|---|---|---|---|---|---|---|---|---|---|---|---|---|
| CLIP (OpenAI) [7] | 99.4 | 83.5 | 93.0 | 95.0 | 74.6 | 74.3 | 75.0 | 44.5 | 67.6 | 57.5 | 65.2 | 88.1 | 71.5 |
| Linear + Last [10] | 75.6 | 37.9 | 14.8 | 85.5 | 34.0 | 9.9 | 21.5 | 9.9 | 19.6 | 19.9 | 19.9 | 32.5 | 22.1 |
| Linear + Similar | 79.7 | 39.5 | 14.6 | 89.0 | 33.1 | 10.5 | 20.4 | 10.0 | 19.9 | 22.4 | 20.3 | 35.3 | 24.0 |
| Linear + Similar + $\mathcal{R}_S$ | 92.6 | 56.0 | 30.6 | 96.3 | 51.3 | 24.7 | 41.8 | 20.4 | 37.9 | 29.2 | 36.5 | 65.8 | 53.7 |
| MLP + Last [9] | 76.6 | 38.2 | 15.6 | 79.2 | 35.3 | 10.6 | 27.7 | 10.0 | 18.1 | 25.4 | 25.0 | 31.6 | 20.3 |
| MLP + Similar | 84.0 | 38.8 | 17.1 | 81.5 | 34.5 | 11.4 | 25.7 | 10.9 | 20.1 | 22.2 | 21.3 | 36.4 | 25.0 |
| MLP + Similar + $\mathcal{R}_S$ | 92.7 | 56.0 | 30.5 | 96.3 | 52.1 | 25.1 | 42.3 | 20.8 | 38.3 | 25.0 | 34.7 | 65.9 | 53.8 |
| CSA + Last [29] | 77.9 | 31.4 | 29.3 | 78.5 | 47.4 | 23.2 | 35.5 | 18.4 | 33.4 | 26.4 | 28.3 | 47.0 | 38.3 |
| CSA + Similar | 80.0 | 33.6 | 28.0 | 80.8 | 47.4 | 23.3 | 34.4 | 17.9 | 33.6 | 22.5 | 29.6 | 48.6 | 39.0 |
| CSA + Similar + $\mathcal{R}_S$ | 91.7 | 61.5 | 28.6 | 97.2 | 56.4 | 26.8 | 44.0 | 21.6 | 37.6 | 28.4 | 38.8 | 56.1 | 43.1 |
| FuseMix + Last [22] | 81.8 | 37.2 | 16.4 | 86.6 | 34.6 | 10.0 | 23.1 | 9.9 | 17.7 | 16.0 | 23.9 | 32.5 | 21.9 |
| FuseMix + Similar | 81.8 | 37.2 | 16.4 | 86.6 | 34.6 | 10.0 | 23.1 | 9.9 | 17.7 | 16.0 | 23.9 | 32.5 | 21.9 |
| FuseMix + Similar + $\mathcal{R}_S$ | 91.7 | 55.8 | 29.3 | 96.2 | 46.3 | 21.1 | 35.6 | 19.1 | 34.4 | 27.4 | 36.6 | 58.2 | 47.6 |

| Method | Gtsrb | Kitti | Country211 | FER2013 | PCam | Cars | Aircraft | Pets | Flowers | MNIST | DTD | COCO I2T | COCO T2I |
|---|---|---|---|---|---|---|---|---|---|---|---|---|---|
| CLIP (OpenAI) [7] | 45.3 | 25.2 | 28.8 | 52.5 | 61.5 | 50.96 | 32.2 | 92.7 | 75.2 | 59.7 | 52.4 | 34.6 | 18.5 |
| Linear + Last [10] | 3.4 | 26.8 | 1.4 | 19.7 | 51.2 | 1.0 | 1.3 | 7.0 | 0.2 | 13.7 | 9.0 | 6.9 | 3.6 |
| Linear + Similar | 4.0 | 29.5 | 1.1 | 21.9 | 53.0 | 0.9 | 1.5 | 4.9 | 0.1 | 7.9 | 8.2 | 7.1 | 4.0 |
| Linear + Similar + $\mathcal{R}_S$ | 7.8 | 28.5 | 2.3 | 30.9 | 43.8 | 2.5 | 2.6 | 13.2 | 6.2 | 12.7 | 18.4 | 18.2 | 13.3 |
| MLP + Last [9] | 7.8 | 24.1 | 1.2 | 18.9 | 50.6 | 0.9 | 2.1 | 5.3 | 0.5 | 14.4 | 8.4 | 6.0 | 2.7 |
| MLP + Similar | 7.4 | 25.4 | 1.3 | 17.9 | 50.3 | 1.2 | 1.6 | 6.1 | 0.4 | 9.9 | 9.4 | 7.1 | 3.8 |
| MLP + Similar + $\mathcal{R}_S$ | 7.9 | 32.7 | 2.5 | 28.2 | 47.0 | 2.2 | 3.3 | 15.3 | 5.6 | 13.6 | 18.7 | 18.3 | 13.4 |
| CSA + Last [29] | 7.0 | 27.1 | 1.6 | 28.1 | 42.9 | 1.8 | 1.5 | 14.4 | 6.2 | 10.2 | 12.6 | 9.2 | 8.0 |
| CSA + Similar | 6.0 | 22.3 | 1.7 | 27.6 | 51.4 | 1.7 | 1.9 | 14.9 | 4.9 | 11.6 | 13.0 | 9.6 | 8.1 |
| CSA + Similar + $\mathcal{R}_S$ | 4.9 | 31.6 | 2.2 | 30.9 | 45.0 | 1.7 | 2.1 | 17.0 | 4.1 | 11.8 | 14.8 | 10.3 | 8.1 |
| FuseMix + Last [22] | 3.3 | 32.0 | 1.1 | 22.5 | 52.3 | 1.2 | 1.4 | 8.0 | 2.0 | 10.7 | 9.9 | 5.8 | 4.0 |
| FuseMix + Similar | 3.3 | 32.0 | 1.1 | 22.5 | 52.3 | 1.2 | 1.4 | 8.0 | 2.0 | 10.7 | 9.9 | 5.8 | 4.0 |
| FuseMix + Similar + $\mathcal{R}_S$ | 5.3 | 28.9 | 2.0 | 24.3 | 50.2 | 1.5 | 1.4 | 11.3 | 2.9 | 15.0 | 15.7 | 15.0 | 10.6 |

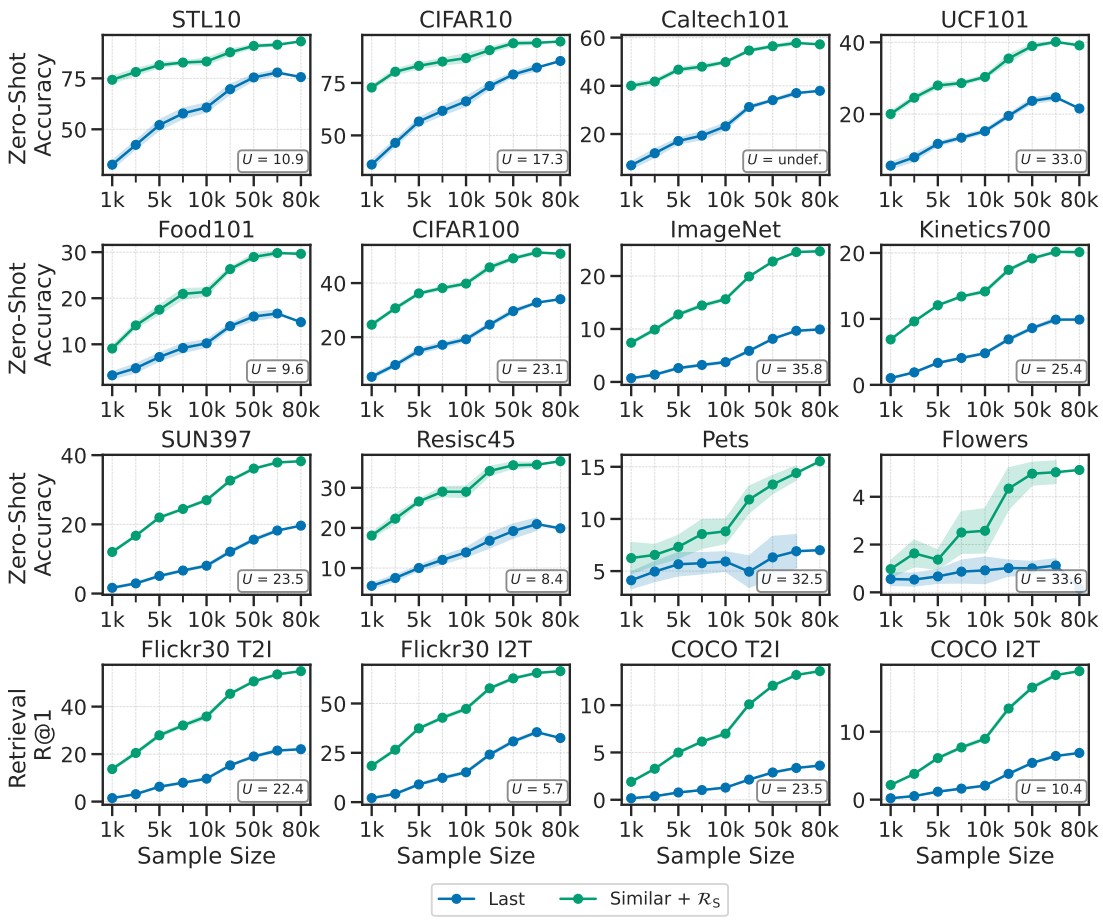

Figure 11: Zero-shot and retrieval performance for training linear alignment using the last or most similar layer. Here, $U$ quantifies the proposed method's label efficiency by computing the utility compared to using the last layer. This figure expands on Figure 3 with more evaluation tasks.

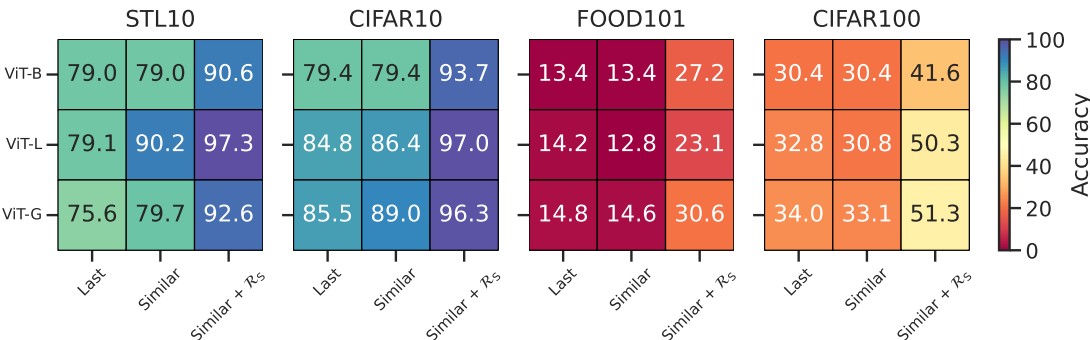

Figure 12: Performance of different vision model combinations when choosing the last or most similar layer for alignment and the influence of adding STRUCTURE regularization when training a single linear alignment layer.

Table 12: Performance for different model combinations for linear alignment when aligning the last or the most similar layer and when adding $\mathcal{R}_S$. This table expands on Figure 7 from the main paper, detailing results on 22 zero-shot classification and two retrieval datasets.

| | | | Zero-shot Classification (Accuracy) | | | | | | | | | | | Retrieval (R@1) | |
|---|---|---|---|---|---|---|---|---|---|---|---|---|---|---|---|
| Language Model | Vision Model | Method | STL10 | Caltech101 | Food101 | CIFAR10 | CIFAR100 | ImageNet | UCF101 | Kinetics700 | SUN397 | EuroSAT | Resisc45 | Flickr30 I2T | Flickr30 T2I |
| RoBERTa | ViT-L | Last [10] | 79.1 | 34.2 | 14.2 | 84.8 | 32.8 | 8.9 | 21.9 | 9.0 | 16.1 | 12.8 | 23.2 | 28.2 | 19.7 |
| | | Similar | 90.2 | 39.8 | 12.8 | 86.4 | 30.8 | 10.3 | 24.2 | 8.4 | 18.4 | 16.4 | 15.0 | 36.3 | 24.3 |
| | | Similar + $\mathcal{R}_S$ | 97.3 | 60.4 | 23.1 | 97.0 | 50.3 | 22.7 | 33.9 | 16.8 | 35.5 | 25.8 | 26.2 | 58.8 | 47.4 |
| | ViT-G | Last [10] | 75.6 | 37.9 | 14.8 | 85.5 | 34.0 | 9.9 | 21.5 | 9.9 | 19.6 | 19.9 | 19.9 | 32.5 | 22.1 |
| | | Similar | 79.7 | 39.5 | 14.6 | 89.0 | 33.1 | 10.5 | 20.4 | 10.0 | 19.9 | 22.4 | 20.3 | 35.3 | 24.0 |
| | | Similar + $\mathcal{R}_S$ | 92.6 | 56.0 | 30.6 | 96.3 | 51.3 | 24.7 | 41.8 | 20.4 | 37.9 | 29.2 | 36.5 | 65.8 | 53.7 |
| Llama3-8B | ViT-G | Last [10] | 68.9 | 27.5 | 22.7 | 74.3 | 27.0 | 6.1 | 12.8 | 5.7 | 15.5 | 13.9 | 12.4 | 21.8 | 11.5 |
| | | Similar | 85.3 | 24.4 | 26.5 | 80.8 | 29.4 | 7.3 | 17.2 | 6.7 | 18.6 | 8.5 | 11.1 | 30.7 | 21.5 |
| | | Similar + $\mathcal{R}_S$ | 91.3 | 53.9 | 28.8 | 89.4 | 41.1 | 17.6 | 32.5 | 14.0 | 31.7 | 24.6 | 29.7 | 56.1 | 44.0 |
| Llama13B | ViT-G | Last [10] | 70.8 | 33.5 | 20.6 | 69.5 | 26.4 | 5.4 | 8.6 | 5.7 | 12.7 | 20.7 | 13.0 | 22.3 | 12.4 |
| | | Similar | 79.2 | 31.9 | 18.1 | 69.9 | 23.1 | 4.9 | 13.9 | 5.2 | 14.2 | 11.2 | 9.9 | 26.3 | 17.1 |
| | | Similar + $\mathcal{R}_S$ | 93.6 | 59.7 | 36.5 | 87.3 | 48.4 | 21.5 | 42.7 | 15.8 | 32.6 | 23.4 | 26.2 | 56.7 | 48.5 |
| BGE-Small | ViT-G | Last [10] | 75.9 | 39.0 | 14.5 | 82.7 | 30.4 | 7.5 | 24.4 | 8.0 | 14.6 | 14.2 | 18.1 | 31.6 | 19.1 |
| | | Similar | 81.9 | 37.8 | 14.2 | 87.3 | 29.5 | 8.2 | 25.7 | 7.3 | 16.1 | 12.7 | 15.2 | 35.9 | 23.0 |
| | | Similar + $\mathcal{R}_S$ | 91.3 | 52.2 | 22.1 | 92.8 | 44.8 | 17.2 | 35.3 | 14.9 | 25.8 | 30.5 | 29.4 | 56.9 | 45.6 |
| BGE-Base | ViT-G | Last [10] | 82.5 | 36.7 | 16.5 | 77.2 | 36.0 | 9.4 | 28.3 | 9.4 | 17.2 | 23.7 | 21.4 | 31.7 | 18.0 |
| | | Similar | 81.2 | 38.6 | 15.3 | 77.1 | 34.1 | 9.3 | 27.9 | 9.2 | 17.7 | 14.2 | 19.4 | 32.2 | 20.9 |
| | | Similar + $\mathcal{R}_S$ | 87.0 | 54.7 | 23.4 | 89.7 | 46.7 | 20.8 | 42.6 | 17.7 | 34.8 | 25.9 | 34.3 | 60.0 | 48.0 |
| all-MiniLM-L6-v2 | ViT-T | Last [10] | 86.6 | 40.8 | 5.2 | 66.0 | 17.3 | 5.3 | 12.1 | 3.6 | 9.4 | 22.5 | 7.4 | 16.9 | 14.1 |
| | | Similar | 86.6 | 40.8 | 5.2 | 66.0 | 17.3 | 5.3 | 12.1 | 3.6 | 9.4 | 22.5 | 7.4 | 16.9 | 14.1 |
| | | Similar + $\mathcal{R}_S$ | 89.2 | 45.6 | 6.7 | 68.9 | 23.5 | 8.9 | 17.7 | 6.0 | 13.8 | 21.9 | 9.9 | 23.6 | 17.9 |
| all-mpnet-base-v2 | ViT-T | Last [10] | 85.7 | 42.9 | 5.9 | 65.2 | 22.9 | 8.9 | 18.5 | 5.9 | 16.0 | 18.7 | 9.3 | 22.6 | 14.8 |
| | | Similar | 85.7 | 42.9 | 5.9 | 65.2 | 22.9 | 8.9 | 18.5 | 5.9 | 16.0 | 18.7 | 9.3 | 22.6 | 14.8 |
| | | Similar + $\mathcal{R}_S$ | 85.1 | 52.1 | 8.7 | 68.0 | 28.2 | 12.1 | 21.6 | 7.8 | 19.7 | 26.4 | 12.6 | 28.9 | 22.3 |

| | | | Zero-shot Classification (Accuracy) | | | | | | | | | | | Retrieval (R@1) | |
|---|---|---|---|---|---|---|---|---|---|---|---|---|---|---|---|
| Language Model | Vision Model | Method | Gtsrb | Kitti | Country211 | FER2013 | PCam | Cars | Aircraft | Pets | Flowers | MNIST | DTD | COCO I2T | COCO T2I |
| RoBERTa | ViT-L | Last [10] | 3.3 | 30.2 | 1.0 | 22.4 | 50.8 | 1.3 | 1.4 | 7.9 | 2.0 | 10.9 | 9.7 | 5.1 | 3.4 |
| | | Similar | 4.2 | 23.7 | 1.0 | 22.2 | 50.2 | 1.4 | 1.0 | 10.3 | 1.4 | 11.6 | 10.0 | 6.1 | 3.9 |
| | | Similar + $\mathcal{R}_S$ | 4.6 | 28.2 | 2.0 | 25.3 | 50.0 | 1.9 | 3.1 | 17.5 | 6.5 | 11.6 | 17.1 | 15.5 | 10.6 |
| | ViT-G | Last [10] | 3.4 | 26.8 | 1.4 | 19.7 | 51.2 | 1.0 | 1.3 | 7.0 | 0.2 | 13.7 | 9.0 | 6.9 | 3.6 |
| | | Similar | 4.0 | 29.5 | 1.1 | 21.9 | 53.0 | 0.9 | 1.5 | 4.9 | 0.1 | 7.9 | 8.2 | 7.1 | 4.0 |
| | | Similar + $\mathcal{R}_S$ | 7.8 | 28.5 | 2.3 | 30.9 | 43.8 | 2.5 | 2.6 | 13.2 | 6.2 | 12.7 | 18.4 | 18.2 | 13.3 |
| Llama3-8B | ViT-G | Last [10] | 5.0 | 27.8 | 0.7 | 18.0 | 54.6 | 0.4 | 2.2 | 3.2 | 3.4 | 13.2 | 4.9 | 3.5 | 2.8 |
| | | Similar | 2.7 | 26.1 | 0.9 | 23.8 | 47.4 | 1.4 | 4.3 | 10.6 | 5.6 | 7.7 | 11.9 | 5.4 | 5.2 |
| | | Similar + $\mathcal{R}_S$ | 3.4 | 28.9 | 2.1 | 19.3 | 53.8 | 1.8 | 8.0 | 15.9 | 10.0 | 14.5 | 17.9 | 11.8 | 10.5 |
| Llama13B | ViT-G | Last [10] | 4.5 | 37.4 | 1.0 | 13.7 | 58.2 | 1.9 | 2.7 | 6.7 | 1.9 | 14.5 | 6.3 | 4.0 | 3.0 |
| | | Similar | 5.7 | 35.7 | 0.7 | 13.4 | 49.6 | 1.8 | 2.8 | 7.3 | 3.4 | 12.4 | 8.6 | 4.9 | 4.1 |
| | | Similar + $\mathcal{R}_S$ | 9.9 | 33.5 | 1.6 | 30.0 | 52.2 | 2.8 | 6.6 | 19.9 | 7.8 | 13.0 | 17.3 | 13.9 | 12.8 |
| BGE-Small | ViT-G | Last [10] | 5.9 | 26.4 | 0.9 | 21.5 | 50.3 | 0.7 | 0.8 | 6.3 | 1.8 | 8.0 | 8.6 | 6.0 | 3.2 |
| | | Similar | 4.2 | 25.5 | 1.0 | 19.5 | 51.2 | 0.9 | 1.1 | 9.5 | 5.2 | 4.6 | 8.5 | 7.1 | 4.4 |
| | | Similar + $\mathcal{R}_S$ | 7.4 | 40.7 | 1.5 | 36.9 | 50.4 | 1.6 | 1.4 | 12.2 | 6.1 | 9.8 | 12.5 | 12.8 | 10.3 |
| BGE-Base | ViT-G | Last [10] | 6.6 | 36.2 | 0.9 | 20.6 | 50.3 | 0.9 | 2.0 | 7.5 | 1.6 | 11.7 | 5.6 | 5.5 | 2.7 |
| | | Similar | 6.2 | 35.5 | 0.9 | 22.9 | 50.2 | 0.7 | 2.5 | 7.9 | 2.6 | 9.0 | 7.3 | 6.0 | 3.7 |
| | | Similar + $\mathcal{R}_S$ | 2.9 | 42.8 | 2.2 | 29.9 | 50.0 | 1.5 | 2.3 | 10.0 | 3.9 | 11.0 | 15.1 | 15.2 | 11.4 |
| all-MiniLM-L6-v2 | ViT-T | Last [10] | 5.6 | 11.7 | 0.7 | 23.1 | 50.1 | 0.3 | 1.2 | 7.8 | 3.5 | 9.3 | 5.7 | 2.8 | 2.5 |
| | | Similar | 5.6 | 11.7 | 0.7 | 23.1 | 50.1 | 0.3 | 1.2 | 7.8 | 3.5 | 9.3 | 5.7 | 2.8 | 2.5 |
| | | Similar + $\mathcal{R}_S$ | 3.5 | 14.7 | 1.0 | 13.3 | 48.3 | 0.9 | 1.3 | 6.4 | 0.8 | 9.6 | 6.3 | 4.5 | 3.2 |
| all-mpnet-base-v2 | ViT-T | Last [10] | 1.4 | 26.7 | 1.1 | 14.8 | 45.2 | 1.1 | 2.4 | 5.3 | 1.1 | 9.4 | 10.2 | 4.3 | 2.9 |
| | | Similar | 1.4 | 26.7 | 1.1 | 14.8 | 45.2 | 1.1 | 2.4 | 5.3 | 1.1 | 9.4 | 10.2 | 4.3 | 2.9 |
| | | Similar + $\mathcal{R}_S$ | 2.8 | 23.0 | 1.4 | 14.9 | 45.3 | 1.2 | 1.5 | 6.1 | 1.0 | 10.6 | 12.1 | 6.3 | 4.5 |

