# OpenReview forum: "With Limited Data for Multimodal Alignment, Let the STRUCTURE Guide You"
_NeurIPS.cc/2025/Conference — NeurIPS 2025 poster_

### Official Review · Reviewer_y6jJ · 2025-06-11

**Clarity:** 3
**Significance:** 3
**Originality:** 3
**Rating:** 5
**Confidence:** 4

**Summary:**

This paper presents STRUCTURE, a novel regularization technique that preserves the neighborhood structure of the latent space in pretrained unimodal encoders. It enables more effective and efficient multimodal alignment, especially in low-data training scenarios. The technique can be added to existing alignment methods and helps maintain the geometric relationships in the input representations. The authors show that their method performs well with significantly less training data, resulting in better alignment and improved performance in downstream tasks.

**Questions:**

Questions:
- The definition of hierarchical level L in Eq. (4) is not intuitive. Could the authors give more insight or examples to help understand its role?
- Have the authors tested STRUCTURE with smaller, less powerful networks? This would help understand whether the method still works when the pretrained representations are less aligned.
- The paper [A] ("Mind the Gap", NeurIPS 2022) highlights the modality gap in contrastive models like CLIP. Since STRUCTURE is used with alignment losses (often contrastive), is this gap still present in the shared embedding space $\mathcal{A}$? Does STRUCTURE help reduce the modality gap, or does it preserve it? How does preserving structure interact with or influence this phenomenon?

[A] Mind the Gap: Understanding the Modality Gap in Multi-modal Contrastive Representation Learning. NeuIPS 2022.

**Ethical Concerns:**

["NO or VERY MINOR ethics concerns only"]

**Final Justification:**

After carefully reviewing the authors' rebuttal and considering the discussion, I am satisfied that most of my concerns have been adequately addressed.

In particular:

1) The additional experiments with smaller networks.
2) The explanation of the hierarchical level


The authors’ responsiveness and thoroughness in addressing each point, including new experiments across modalities and datasets, demonstrate the robustness and generality of their method.

Overall, I find the work technically sound, relevant, and well-executed. I am raising my score accordingly.

**Limitations:**

yes

**Quality:**

3

**Strengths And Weaknesses:**

Strengths:
- Requires less training data to achieve strong alignment, making it suitable for low-resource settings.
- As a regularization method, it can be integrated into many existing alignment frameworks.
- The paper includes a strong set of experiments across a wide range of benchmarks.

Weaknesses:
- The unimodal models used (RoBerta and DinoV2 Giant) are very deep and computationally expensive. STRUCTURE shows more benefits when aligning highly similar layers, but these benefits may decrease when using smaller or weaker models (simple Resnets or base ViT).
- The proposed framework is only tested on two modalities. There is no discussion or evidence on how it would extend to three or more modalities. Moreover within the two modal case, just text and image modalities are tested.
- It's not clear whether the variable N in Eq. (2) and lines 137–138 refers to the size of the whole dataset (as suggested in line 105) or just the batch size during training. If it's the batch size, the similarity matrix may only capture local (within-batch) relationships, which could limit global alignment quality.

---

> ### Author Rebuttal · Authors · 2025-07-29
>
> We would like to extend our sincere appreciation to the reviewer for their insightful comments and positive evaluation of our work. We are encouraged by the recognition that our approach requires less training data to achieve strong alignment, making it particularly well-suited to low-resource settings. We are likewise gratified that the method’s versatility as a regularization technique, enabling seamless integration into existing alignment frameworks, was noted. Finally, we appreciate the acknowledgement of our comprehensive experimental suite, which spans a wide range of benchmarks.
>
> > 1.1 & 1.2. STRUCTURE with Smaller Networks
>
> We focus on large models because they are readily available and are not being updated within our framework, making them essentially an initial feature extraction. Additionally, we would like to highlight that Figure 11 of Appendix I.6. (referenced in the main paper on L225) includes experiments with a base ViT.
>
> In response to the Reviewer’s feedback, we conducted additional experiments of two model combinations, which can be considered much smaller. Specifically, in the first table, we align a ViT-Tiny with 5.72 million parameters (even smaller than a ResNet18) and an all-MiniLM-L6-v2 with 22.7 million parameters. In the second table, we align a ViT-Tiny with 5.72 million parameters and an all-mpnet-base-v2 with 109 million parameters. Both combinations are trained using linear alignment functions. Results show that adding STRUCTURE still yields a benefit even for these smaller model combinations.
>
> We would like to highlight two key aspects of these results. First, the alignment score, in terms of mutual kNN, for these pairs of models is significantly lower than for the other (bigger) combinations. Specifically, 0.09 for  DINO\_v2 ViT-Tiny+all-MiniLM-L6-v2 and 0.10 for DINO\_v2 ViT-Tiny+all-mpnet-base-v2, which are much lower than the 0.29 for DINO\_v2 ViT-L+RoBERTa. Second, as the Platonic Representation Hypothesis suggests, the convergence of the models to the same representation increases with scale and performance. Thus, size itself has a significant influence on the relative performance of the aligned models, as more similar models facilitate their alignment, and enforcing hierarchical relationships becomes more meaningful. However, even when the models are less similar and respectively smaller, the proposed regularization yields consistent improvements.
>
> ViT-Tiny+all-MiniLM-L6-v2:
>
> |  | STL10 | CIFAR10 | Caltech101 | Food101 | CIFAR100 | ImageNet | Pets | Flickr I2T | Flickr T2I |
> | :---- | :---: | :---: | :---: | :---: | :---: | :---: | :---: | :---: | :---: |
> | Last | 86.6 | 65.9 | 40.9 | 5.1 | 17.4 | 5.3 | 4.7 | 17.2 | 14.1 |
> | Similar | 86.6 | 65.9 | 40.9 | 5.1 | 17.4 | 5.3 | 4.7 | 17.2 | 14.1 |
> | Similar \+ R\_S | 89.3 | 68.8 | 45.6 | 6.8 | 23.7 | 8.9 | 6.2 | 24.1 | 18.0 |
>
> ViT-Tiny+all-mpnet-base-v2:
>
> |  | STL10 | CIFAR10 | Caltech101 | Food101 | CIFAR100 | ImageNet | Pets | Flickr I2T | Flickr T2I |
> | :---- | :---: | :---: | :---: | :---: | :---: | :---: | :---: | :---: | :---: |
> | Last | 85.8 | 63.9 | 47.6 | 5.7 | 22.8 | 8.9 | 6.0 | 22.7 | 15.5 |
> | Similar | 85.6 | 65.3 | 42.9 | 5.9 | 22.8 | 8.9 | 5.2 | 22.9 | 14.8 |
> | Similar \+ R\_S | 85.1 | 68.0 | 52.1 | 8.8 | 28.1 | 12.2 | 6.0 | 28.7 | 22.1 |
>
> > 2\. STRUCTURE on Three or More Modalities and Additional Two Modal Cases
>
> First, we would like to discuss the extension of the proposed regularization to aligning more than two modalities. As regularization is only calculated on the pretrained representations of each individual modality, thus not relying on any cross-modal data, extending this approach involves computing the regularization in each modality separately and adding an additional term to the loss function. Specifically, Eq. 8 would get a third term for the new modality:
>
> $$
> \mathcal{L} = \mathcal{L}_{\text{A}} + \lambda\left(
> \mathcal{R}_S^{(L)}(X_1, f_1(X_1)) +
> \mathcal{R}_S^{(L)}(X_2, f_2(X_2)) +
> \mathcal{R}_S^{(L)}(X_3, f_3(X_3))
> \right)
> $$
>
> Thus, regardless of the alignment objective function used, the extension of regularization is indeed straightforward and we will add a discussion in our manuscript.
>
> In addition to text and vision modalities, we would like to point out the additional experiments on the audio and text domains in the original submission in Appendix I.3. In addition, in response to the reviewer’s comment, we have now conducted additional experiments on the *biological domain* using only 19,900 paired human single-cell image-transcriptome data. In this experiment, we linearly align a UCE \[1\] encoder for the transcriptome data and a masked autoencoder (MAE) trained on single-cell images for the image data. We split the 19,900 paired samples into 100 training samples and 19,800 test samples to simulate the low data setting. Results below show that our regularization also helps in the biological domain, yielding relative improvements of 43.1% on the retrieval task and 30.3% on the classification task, even if the Platonic representation hypothesis is not guaranteed in the biological domain.
>
> |  | Retrieval | Classification |
> | :---- | :---: | :---: |
> | Last | 4.2 | 40.3 |
> | Last \+ R\_S | 6.0 | 52.5 |
>
> \[1\] Universal Cell Embeddings: A Foundation Model for Cell Biology, bioRxiv 2023
>
> > 3\. N in Eq. (2)
>
> The variable N in the default setting of STRUCTURE itself represents the size of the batch, typically set to around 4,096 random samples (approximately 6% of the entire dataset), which allows for a sufficient estimate of the true data generation distribution while maintaining computational efficiency. In response to the reviewer’s feedback, we conducted experiments to demonstrate how the choice of sample size affects performance. Results are given when aligning RoBERTa with a ViT-G using linear alignment. Results show that the benefit of the regularization increases with increased sample size, but also show that it’s efficiency, since with even 128 samples, there is a clear improvement.
>
> | N | STL10 | CIFAR10 | Caltech101 | Food101 | CIFAR100 | ImageNet | Pets | Flickr I2T |
> | :---- | :---: | :---: | :---: | :---: | :---: | :---: | :---: | :---: |
> | 128 | 90.3 | 96.5 | 55.2 | 29.2 | 48.6 | 22.7 | 12.9 | 60.5 |
> | 1,024 | 92.1 | 96.8 | 54.9 | 30.2 | 50.4 | 24.0 | 13.5 | 64.8 |
> | 4,096 | 92.6 | 96.3 | 56.0 | 30.6 | 51.3 | 24.7 | 13.2 | 65.8 |
> | 8,192 | 92.5 | 96.5 | 56.6 | 31.0 | 52.1  | 25.0 | 13.6 | 66.8 |
>
> > 4\. Understanding of Hierarchical Level L
>
> The hierarchical level $L$ in Equation 4 can be understood through the lens of a random walk on the data manifold. Specifically, the row-normalized similarity matrix, $P\_X$, can be interpreted as the transition matrix for a random walk on a graph where data points are nodes and similarity defines edge weights. In this context, the value $P\_X(i, j)$ represents the probability of transitioning from sample $i$ to sample $j$ in a single step. The hierarchical levels then correspond to the length of this random walk:
>
> (1) $l \= 1$ (1-hop): represents the probability of a direct transition between two samples. Preserving this level enforces consistency in the immediate local neighborhood structure. It ensures that if sample $j$ is a close neighbor of $i$ in the original space, it remains so in the aligned space.
>
> (2) $l \> 1$ (l-hop): the matrix power $(P\_X)^l$ computes the probability of reaching one sample from another in exactly $l$ steps. This captures progressively larger structural patterns. For $l=2$, it considers paths through one intermediary. This enforces consistency at the level of local clusters or "neighborhoods of neighborhoods." Two samples that are not direct neighbors but belong to the same tight cluster will have high 2-hop probabilities. As $l$ increases, we capture more global, long-range connectivity across the data manifold.
>
> By enforcing consistency between the multi-step transition probabilities of the original space $(P\_X)^l$ and the aligned space $(P\_A)^l$, the STRUCTURE regularizer ensures the alignment function preserves the intrinsic geometry of the data manifold at multiple scales. This prevents the "over-warping" that can occur in unregularized alignment, where local neighborhoods might be preserved but the global arrangement of clusters is distorted.
>
> While our experiments show that setting $L=1$ is often sufficient, this is because perfect 1-hop consistency mathematically implies perfect $l$-hop consistency for all $l$. Therefore, strongly regularizing the most local structure provides a powerful, indirect constraint on the entire manifold geometry. Explicitly including higher levels ($L\>1$) simply makes this constraint more direct, which can provide additional benefits on fine-grained tasks where subtle, multi-hop relationships are more critical to preserve.
>
> We will expand the discussion about it in the revised manuscript.
>
> > 5\. About Modality Gap
>
> We measured the modality gap for three different alignment setups for a ViT-Giant and RoBERTa Large, trained on COCO with a linear alignment function. The modality gap was measured for the two image-text datasets considered in our submission (i.e., COCO and Flickr) to ensure that the comparison is influenced solely by the data itself and not by any other artifacts introduced by the evaluation, such as a zero-shot template, as would be necessary for the classification datasets. Interestingly, the results show that the modality gap indeed decreases when first selecting the most similar layers and then further when applying the proposed regularization. This aligns with the findings from the respective paper, that the modality gap indeed has a relationship with downstream performance. We will include these findings in the revised manuscript.
>
> |  | COCO | Flickr |
> | :---- | :---: | :---: |
> | Last | 6.6 | 6.8 |
> | Similar | 6.0 | 6.3 |
> | Similar \+ R\_S | 4.5 | 3.8 |

---

> > ### Comment · Reviewer_y6jJ · 2025-08-03
> >
> > Thanks to the authors for their effort in addressing the questions.
> >
> > **STRUCTURE with Smaller Networks**
> >
> > The results are interesting. While the performance improvement is evident, the relative increasing is smaller than that reported in Table 2. These findings are in line with the reviewer's expectations and relate to the earlier discussion on the alignment of initial encoders. The better the alignment, the more your regularization technique can help preserve the structures learned by pretrained encoders, which in turn benefits alignment.
> >
> > **Understanding of Hierarchical Level L**
> >
> > The claim that this approach “can provide additional benefits on fine-grained tasks where subtle, multi-hop relationships are more critical to preserve” is intriguing. However, it should be supported with empirical evidence.
> >
> > Overall, I thank the authors for this insightful work and for their responsiveness in addressing the questions. Most of my concerns have been resolved. I will raise my score.

---

> > > ### Author Response · Authors · 2025-08-04
> > >
> > > We thank the reviewer for the constructive feedback. We are glad the results aligned with the reviewer's expectations and helped address the reviewer's concerns. We appreciate the positive assessment of our work, as well as the time and engagement the reviewer dedicated to our work.

---

### Official Review · Reviewer_YaRe · 2025-06-15

**Clarity:** 4
**Significance:** 3
**Originality:** 4
**Rating:** 5
**Confidence:** 4

**Summary:**

This paper presents STRUCTURE, a multimodal alignment method that can effectively align two unimodal foundation models into one shared representation space, with only limited labeled (paired) data. The key to STRUCTURE is to enforce consistency between structural relationships (from the original unimodal space to the common space). And the authors further show that selecting the similarity-based layer is better for alignment than simply selecting the last layer.

**Questions:**

Please see weaknesses.  I will raise the score if the concerns about hierarchical relations are resolved.

**Ethical Concerns:**

["NO or VERY MINOR ethics concerns only"]

**Final Justification:**

The authors have addressed my concerns in the rebuttal. Therefore, I would recommend the acceptance of the paper.

**Limitations:**

Yes.

**Paper Formatting Concerns:**

Not found.

**Quality:**

3

**Strengths And Weaknesses:**

Strengths:

-This paper is well-written.

-The idea of preserving hierarchical relationships for alignment is technically sound and interesting.

-The proposed regularization term and layer selection strategy can be seamlessly integrated into other methods to enhance the alignment.

-The results are strong and comprehensive.

Weaknesses:

-The proposed regularization uses $l$-hop reachability to calculate the similarity relation for different hierarchical levels. It seems that introducing such hierarchical relationships is a main contribution of the regularization. However,  as shown in Table 3 and Table 7, integrating the information from multiple hierarchies seems unnecessary. And the hierarchical level $l$ is set to 1 in practice, where only the sample-level relation is considered. The authors should discuss how to effectively use the hierarchical relation. Ideally, the alignment effect would increase as $l$ increases and gradually converge.

-It is also suggested to discuss $l$-hop when first introduced, which would make it easier for readers to understand.

---

> ### Author Rebuttal · Authors · 2025-07-29
>
> We would like to extend our sincere appreciation to the reviewer for their positive assessment and their feedback. We are pleased that the manuscript’s clarity and rigor were noted, and that the reviewer found the idea of preserving hierarchical relationships for alignment technically sound and compelling our results strong and the proposed regularization effective and modular. We thank the reviewer for being open to increasing their score if their concerns are addressed.
>
> > 1\. The authors should discuss how to effectively use the hierarchical relation. Ideally, the alignment effect would increase as l increases and gradually converge.
>
> Regarding the contribution, we would like to emphasize that the main contribution is broader than hierarchical relationships imposed in the regularization and involves the proposition of a regularization strategy that acts as a geometric prior inferred from the pretrained encoder that can enhance different alignment strategies in low data regime.
>
> In the original version, we investigated the effect of increasing l on coarse-grained datasets (STL10, CIFAR10, Food101, and CIFAR100). The results are presented in Table 7 of the Appendix. There, we found that for some coarse-grained datasets, such as CIFAR10 or STL, the influence of the number of hops considered is less impactful. In response to the Reviewer’s question, we have now conducted more comprehensive experiments on additional datasets for different levels in the table below. On datasets that can be regarded as more fine-grained, such as Pets or Flickr, we find that enforcing higher-order interactions has a positive impact and slightly improves the performance.
>
> It is important to note that even if one only considers a single hierarchical level, the modal still enforces consistency between higher-order relationships, as a perfect similarity graph on the lowest level also yields perfect relationships on all higher levels. Thus, increasing the number of hops is a more direct way of enforcing these higher relationships, but the lower levels still enforce them indirectly.
>
> |  | Pets | MNIST | Food101 | UCF101 | Flickr I2T | Flickr T2I | COCO I2T | COCO T2I |
> | :---- | :---: | :---: | :---: | :---: | :---: | :---: | :---: | :---: |
> | $R\_S^1$ | 15.3 | 13.6 | 30.5 | 42.3 | 65.9  | 53.8  | 18.3 | 13.4 |
> | $R\_S^2$ | 15.4 | 13.5 | 30.4 | 42.3 | 66.0  | 53.7 | 18.4 | 13.4 |
> | $R\_S^3$ | 15.8 | 13.6 | 30.4 | 42.2 | 66.2 | 54.0 | 18.5  | 13.5 |
> | $R\_S^5$ | 15.9 | 13.8 | 30.6 | 42.4 | 66.7 | 54.3 | 18.7  | 13.6 |
>
> > 2\. It is also suggested to discuss l\-hop when first introduced, which would make it easier for readers to understand.
>
> We will thus revise paragraphs L143-144 and include a more intuitive description of the l-hops. Specifically, we will add the following sentences after L144: “Intuitively, the similarity graph can be thought of as a Markov chain where the hops refer to the transitions between different states. Thus, the $l$ hops represent the probability of transitions reachable with $l$ hops.” Additionally, we will discuss l-hop in more detail when it is first introduced.

---

> > ### Comment · Reviewer_YaRe · 2025-08-05
> >
> > The authors have addressed my initial concerns in the rebuttal. Therefore, I would like to increase my rating.

---

> > > ### Author Response · Authors · 2025-08-06
> > >
> > > We thank the reviewer for the thoughtful and constructive feedback.
> > > We are pleased that our results met the reviewer’s expectations and effectively addressed the concerns raised.
> > > We appreciate the positive evaluation of our work, as well as the time and careful consideration the reviewer devoted to our submission.

---

### Official Review · Reviewer_u1eZ · 2025-06-22

**Clarity:** 3
**Significance:** 3
**Originality:** 3
**Rating:** 5
**Confidence:** 4

**Summary:**

The authors propose a regularization scheme for multimodal alignment on top of frozen pre-trained unimodal encoders, fit for settings where multimodal data is scarce. The authors also propose a method to select the layer of the unimodal encoders with which to perform multimodal alignment.

**Questions:**

- How are the k samples per class in Figure 5 incorporated into the training set? Are they oversampled or just become another element in the dataset?
- Have the authors considered using text embedding models instead of RoBERTa or LLMs, as was done in FuseMix?

Also see weaknesses above with corresponding questions. Should my concerns and questions be adequately addressed, I will consider increasing my score.

**Ethical Concerns:**

["NO or VERY MINOR ethics concerns only"]

**Final Justification:**

The authors have addressed my concerns and remaining questions to my satisfaction; they promised to update their manuscript accordingly. I will thus recommend acceptance.

**Limitations:**

The authors do not acknowledge limitations or potential negative societal impact of their work.

**Quality:**

3

**Strengths And Weaknesses:**

## Strengths
- The empirical results are strong and convincingly demonstrate the efficacy of the proposed regularization scheme and layer-selection strategy. The authors additionally provide good supporting analyses and ablations to further corroborate their claims.
- The proposed regularizing loss seems novel and is well-motivated from the lens of maintaining the semantic structure of the latent space of the underlying unimodal encoders. The layer-selection strategy is also interesting and novel, and Figure 2 does a good job of motivating the need for such a strategy.
- The paper is well written with a clear methodological description and motivation for design choices.

## Weaknesses
- This paper is very similar in spirit to FuseMix [1] from CVPR 2024 but it was never mentioned in this work: the motivation of using pre-trained and frozen unimodal encoders to combat the scarcity of multimodal data was already explored in FuseMix, even with tens of thousands of multimodal pairs. FuseMix also has a similar setup in terms of proposing a regularization scheme (via a mixup augmentation routine). I acknowledge that there are still differences between this work and FuseMix in terms of the proposed methodology, but some of the key claims and motivations of this paper are not novel so I would ask the authors to include a discussion of their work compared to FuseMix (both in their rebuttal and in the updated manuscript) and include it as a baseline with experimental results. Some of the paper's claims are overstated.
- The authors do not discuss limitations of their proposed method.

## References
[1] Vouitsis, Noël, et al. "Data-efficient multimodal fusion on a single gpu." Proceedings of the IEEE/CVF Conference on Computer Vision and Pattern Recognition. 2024.

---

> ### Author Rebuttal · Authors · 2025-07-29
>
> We would like to extend our sincere appreciation to the reviewer for their thorough and encouraging evaluation. We are grateful by the reviewer’s recognition of the well written manuscript and motivation of our work, the novelty of our methodology, and strong empirical results which convincingly demonstrate the efficacy of our methodology.  We appreciate the reviewer’s willingness to increase their score, provided that our response addresses their concerns and questions.
>
> > 1\. This paper is very similar in spirit to FuseMix \[1\] from CVPR 2024 but it was never mentioned in this work \[...\].
>
> As the reviewer noted, *although FuseMix shared a similar motivation, its underlying methodology differs from our proposed approach*, especially in terms of where the contribution is focused. FuseMix aims to improve the training efficiency by using latent space augmentations. In contrast, our framework suggests selecting the most similar layers for alignment and introduces hierarchical preservation regularization to improve performance in data-limited settings. Thus, **the two methods are complementary to each other** and can be used jointly.
>
> In response to the reviewer’s feedback, we conducted additional experiments: (1) comparing FuseMix to linear alignment enhanced with our methodology (layer selection \+ STRUCTURE regularization R\_S​); and (2) enhancing FuseMix itself by incorporating our methodology into FuseMix to show complementarity. The results show that (1) the proposed framework (Similar \+ R\_S) yields significantly better results than the FuseMix approach in the same setting, specifically leading to an average relative improvement of 64.2% on the considered datasets. Furthermore, the results show that (2) when integrating parts of the proposed framework, such as the layer selection and the regularization, with FuseMix, there is a 64.6% relative improvement compared to the original FuseMix approach. This further demonstrates the flexibility and wide applicability of STRUCTURE.  We will add FuseMix results in our manuscript and discuss it in the introduction and related work section.
>
> |  | STL10 | CIFAR10 | Caltech101 | Food101 | CIFAR100 | ImageNet | Pets | Flickr I2T | Flickr T2I |
> | :---- | :---: | :---: | :---: | :---: | :---: | :---: | :---: | :---: | :---: |
> | Last | 75.6  | 85.5 | 37.9  | 14.8  | 34.0  | 9.9  | 7.0  | 32.5  | 22.1 |
> | Similar | 79.7  | 89.0  | 39.5  | 14.6  | 33.1  | 10.5 | 4.9  | 35.3  | 24.0 |
> | Similar \+ R\_S | 92.6 | 96.3 | 56.0 | 30.6 | 51.3 | 24.7 | 13.2 | 65.8 | 53.7 |
> | FuseMix \+ Last | 77.2 | 85.9 | 39.4 | 16.0 | 35.6 | 11.3 | 9.0 | 35.8 | 24.5 |
> | FuseMix \+ Similar | 81.3 | 89.5 | 40.4 | 15.5 | 35.0 | 11.5 | 6.0 | 37.7 | 26.7 |
> | FuseMix \+ Similar \+ R\_S | 92.1 | 95.9 | 57.1 | 30.4 | 51.5 | 24.9 | 13.5 | 65.3 | 53.5 |
>
> > 2\. The authors do not acknowledge limitations or potential negative societal impact of their work.
>
> We would like to point the reviewer to Appendix A of the original submission, which includes a discussion of the limitations. We did not include negative societal impacts, as the outlined methodology enables multimodal alignment even in limited data scenarios, while being computationally efficient. Thus, we are not aware of any negative societal impacts that would directly result from this work, as noted in L823-L825. To highlight these sections, we will include a reference to them in the main paper.

---

> > ### Comment · Reviewer_u1eZ · 2025-08-02
> >
> > Thank you to the authors for responding to my concerns. However, my two listed questions were not answered. Should the authors include discussions and results around FuseMix in their updated manuscript, move the Limitations section to the main body of the paper (it is an important section that should not be part of the Appendix), and also adequately answer my outstanding questions, then I will increase my score to a 5. It is worth pointing out that the authors’ mentioned limitation “A promising future direction is to systematically investigate how different types of data influence cross-modal alignment quality in a low-data regime” was also investigated in FuseMix, so I would encourage the authors to consider this in future work.

---

> > > ### Author Response · Authors · 2025-08-03
> > >
> > > We will add these new results and discussion about FuseMix in the final version of our manuscript. We will also discuss limitations in the main paper and modify the future work section. We thank the reviewer for the suggestions that helped to further strengthen our work.
> > >
> > > Below are the answers to the two questions:
> > >
> > > >1\. How are the k samples per class in Figure 5 incorporated into the training set? Are they oversampled or just become another element in the dataset?
> > >
> > > The k samples per class are added to the training set. Specifically, to the COCO dataset consisting of 80,000 samples, we add k in-distribution samples per class of a given dataset. For example, for the CIFAR 10 dataset consisting of 10 classes we add 10\*k samples and the final dataset consists of 80,000+10\*k training samples. We will specify this better in the final version of the manuscript.
> > >
> > > >2\. Have the authors considered using text embedding models instead of RoBERTa or LLMs, as was done in FuseMix?
> > >
> > > FuseMix uses E5 and BGE. We did not test these two, but we tested different-sized language models, of which the smallest one (i.e., all-MiniLM-L6-v2) is even smaller than E5 and BGE in terms of the number of parameters (23M parameters), and many are similar in size (i.e., RoBERTa, all-mpnet-base-v2, with around 100M parameters). We, however, agree with the reviewer that adding experiments with pure text embedding methods is an interesting investigation. In response,  we added experiments with BGE-small and BGE-base, and the results are in the table below. Results show that even for these text embeddings the proposed framework yields a significant performance increase. We will add these results in the final version of the manuscript.
> > >
> > > DINO v_2 ViT-Giant+BGE-Small:
> > >
> > > |  | STL10 | CIFAR10 | Caltech101 | Food101 | CIFAR100 | ImageNet | Pets | Flickr I2T | Flickr T2I |
> > > | :---- | :---: | :---: | :---: | :---: | :---: | :---: | :---: | :---: | :---: |
> > > | Last | 82.5 | 77.2 | 36.7 | 16.5 | 36.0 | 9.4 | 7.5 | 31.7 | 18.0 |
> > > | Similar | 81.9 | 87.3 | 37.8 | 14.2 | 29.5 | 8.2 | 9.5 | 35.9 | 23.0 |
> > > | Similar \+ R\_S | 89.4 | 93.3 | 53.2 | 21.8 | 44.9 | 17.1 | 11.0 | 58.0 | 45.3 |
> > >
> > > DINO v_2 ViT-Giant+BGE-Base:
> > >
> > > |  | STL10 | CIFAR10 | Caltech101 | Food101 | CIFAR100 | ImageNet | Pets | Flickr I2T | Flickr T2I |
> > > | :---- | :---: | :---: | :---: | :---: | :---: | :---: | :---: | :---: | :---: |
> > > | Last | 82.5 | 77.2 | 36.7 | 16.5 | 36.0 | 9.4 | 7.5 | 31.7 | 18.0 |
> > > | Similar | 81.2 | 77.1 | 38.6 | 15.3 | 34.1 | 9.3 | 7.9 | 32.2 | 20.9 |
> > > | Similar \+ R\_S | 88.1 | 90.8 | 56.2 | 25.1 | 50.7 | 23.3 | 14.1 | 62.6 | 50.9 |

---

### Official Review · Reviewer_MNry · 2025-07-02

**Clarity:** 4
**Significance:** 3
**Originality:** 3
**Rating:** 5
**Confidence:** 5

**Summary:**

This paper introduces a structure-based regularizer applicable on any latent space transformation. The core idea is to keep the geometry (i.e., local neighborhoods) of the space consistent before and after the transformation. The main proposed application is in the context of building a multimodal model out of independently pre-trained and frozen unimodal foundation models. In the experiments, the proposed regularizer is applied independently on each modality-specific space when optimizing a variety of already existing alignment methods, showing that : i) it allows alignment methods to use significantly fewer paired samples, thanks to the induced structural prior; ii) it substantially increases their performance on different tasks (zero-shot classification and retrieval) but also with different backbone pairs and modalities (text-image, text-audio) ; iii) it is robust to the number of datapoints used to compute it; iv) it can handle distribution mismatches with few in-domain samples.

**Questions:**

## Minor comments/questions

1. It took me a while to understand the difference between trustworthiness and continuity when they are introduced. A different phrasing might help. What clarified it for me was thinking of it as a binary classification task with labels "neighbor" and "not neighbor" and the ground truth being the labeling in the original space while the prediction being the new one. This made it easier to relate trustworthiness as capturing "false positives" (precision), while continuity to "false negatives" (recall).
2. I would argue that 1000 labeled samples are "extremely resource-constrained" when compared to the massive scale at which multimodal models like CLIP are trained, not in general. As an example, consider visual alignment of low-resourced languages, or scientific data where labeling requires costly wet-lab experiments.
3. Reference [9] is from CVPR 2025, not 2024.
4. I think the “Train-test data distribution shift.” is undersold. The fact that as few as 10 samples per class enhance the performance this much should be better highlighted! At the same time, I’m wondering if adding a mix of 10 samples for each class in each dataset has a consistent improvement across all datasets, or if specific samples for the task are needed while other specific ones might have a noising effect when added.
5. Why are the scaling up and down different experiments and not unified?
6. Where are the most suitable layers for the alignment found? What’s the similarity distribution?I would find a heatmap comparing each layer of the first model to each other layer of the other model quite interesting and informative to how this matching is helping. If substantial, this analysis could replace one of the two scaling up/down plots if they were merged.
7. Table 2 is already pretty comprehensive, but I was wondering if an extra column could be squeezed in to show the row-wise average and variance. It would help when comparing them.

## Major Questions

1. Are the classification/retrieval errors uniform enough across classes? Are there cases where almost entire classes are consistently misclassified? It would makes sense if they cannot be reliably mapped across spaces or their relative positions with respect to other classes significantly differ across modalities. It might be helpful checking macro/weighted F1 instead of the coarser accuracy, or to analyze the error variance/distribution across classes for each dataset.
2. Is there an intuition behing this specific choice of normalization (i.e., first L2 norm, then centering)? Have you tried different ones like Standard Scaling or first centering and then normalizing? Same for the distance function used.
3. My understanding is that STRUCTURE is a regularization term that acts independently on each space, but is applied on all the spaces to be aligned to preserve their structure in the process. However, the regularization term doesn’t rely on paired data, so I was wondering if the authors could provide tests in a setting where the **paired** training data for the alignment function is different from the **unpaired** training data for the regularizer. This would help in understanding if the structural prior that STRUCTURE maintains is generalizable to the whole original space, or if it is tied to the specific subspace described by the paired samples.
4. Is the STRUCTURE loss always computed on all the data used to train the alignment function? Take as an example the Figure 3 setting, where the (paired) training data is increased to show its robustness. Varying the training data while computing the STRUCTURE regularization only on a fixed subset (even **unpaired, a random subset of the original space,** to combine this analysis with the one described in the previous bullet) would be a stronger signal that STRUCTURE itself needs less data to be effective.

---

While not a breakthrough, the method is elegant, well-motivated, and practically impactful. Combined with the writing clarity, it makes it a clear accept for me. Among the questions in the list, my priority to consider increasing the rating would be a better disentangling and characterization *regarding the needed amount of in-domain paired data between STRUCTURE and the alignment function*.

**Ethical Concerns:**

["NO or VERY MINOR ethics concerns only"]

**Final Justification:**

The authors have provided an excellent rebuttal (I also read through the other discussions), strenghtening an already quite strong submission. Although a clear accept, I believe it doesn't fully meet the bar for a 6. Therefore, I'm happy to confirm my original score of 5 while thanking the authors again for their work.

**Limitations:**

Yes.

**Paper Formatting Concerns:**

N/A.

**Quality:**

4

**Strengths And Weaknesses:**

## Strengths (Quality, Clarity, Significance)

1. The paper is well-rounded. It has a high variety of experiments and ablations that validate the proposed method.
2. I found it extremely easy to follow. From the start and throughout it, it clearly communicates the high-level problem, the proposed approach, why it is relevant, and, at each step/paragraph, what is being done and why. The reader is guided through it. This is due to a mix of strong writing and well-structured presentation.
3. It clearly articulates and concretely shows the advantages of imposing a structural prior when trying to bridge different modalities with a simple and effective method. I believe it has the potential to be widely adopted and/or extended in this subfield.

## Minor Weakness

I believe a core citation/comparison missing is [a], a method that is similar with respect to: i) the task, since it creates CLIP-like models starting from pre-trained and frozen unimodal encoders; ii) the procedure, since it relies on their **similar neighborhood structure.** More in general, I think that some room in the related works should be dedicated to works in representation alignment methods for latent spaces, especially the ones focusing on geometric priors [e.g., b, c, d, e]. I would suggest broadening the “Platonic representation hypothesis.”, reframing it to this more general topic.

---

**[a]** ASIF: Coupled data turns unimodal models to multimodal without training; Antonio Norelli, Marco Fumero, Valentino Maiorca, Luca Moschella, Emanuele Rodolà, Francesco Locatello; NeurIPS 2023.

**[b]** Relative representations enable zero-shot latent space communication; Luca Moschella, Valentino Maiorca, Marco Fumero, Antonio Norelli, Francesco Locatello, Emanuele Rodolà; ICLR 2023.

**[c]** Latent space translation via semantic alignment; Valentino Maiorca, Luca Moschella, Antonio Norelli, Marco Fumero, Francesco Locatello, Emanuele Rodolà; NeurIPS 2023.

**[d]** Offline bilingual word vectors, orthogonal transformations and the inverted softmax; Samuel L. Smith, David H. P. Turban, Steven Hamblin, Nils Y. Hammerla; ICLR 2017.

**[e]** A robust self-learning method for fully unsupervised cross-lingual mappings of word embeddings; Mikel Artetxe, Gorka Labaka, Eneko Agirre; ACL 2018.

---

> ### Author Rebuttal · Authors · 2025-07-29
>
> We thank the reviewer for the positive assessment, detailed review and the insightful questions.  We appreciate reviewer’s recognition of the clear, well-structured presentation and strong writing, the effectiveness and design of our methodology, and the variety and extensiveness of our experiments. We are encouraged by the reviewer’s belief that our method has the potential for wide adoption, and we are grateful for their willingness to reconsider increasing their score based on our response. We will include all suggestions in the final version of our manuscript.
>
> ## Evaluation Setup:
>
> Throughout the response, we linearly align RoBERTa and ViT-G and report accuracy for classification and recall@1 for retrieval, unless otherwise stated.
>
> ## Major questions:
>
> > 1\. Class-wise Errors and Additional Metrics
>
> We chose the same evaluation setup as for CLIP, including benchmarks and evaluation metrics. However, we fully agree that using a metric that takes into account the class distribution is beneficial.
>
> We report (1) the macro F1 score and (2) the standard deviation of the per-class F1 score to analyze the variance across classes. Results in terms of macro F1 score show that the performance, using our proposed framework, brings significant performance gains.
>
> | F1 score | STL10 | CIFAR100 | Food101 | ImageNet |
> | :---- | :---: | :---: | :---: | :---: |
> | Last | 74.9 | 32.6 | 12.6 | 9.7 |
> | Similar | 79.5 | 31.4 | 12.7 | 10.1 |
> | Similar \+ R\_S | 92.2 | 47.3 | 24.0 | 20.8 |
>
> When comparing the standard deviation of the per-class F1 score, we observe that there is indeed a large variation in scores across classes. For CIFAR100, Food101, and ImageNet, the spread increases when using STRUCTURE, which is expected since the classes represented in the alignment dataset (here, COCO) will experience a performance increase, whereas the others will likely remain the same due to the lack of coverage.
>
> | Stdev $\\downarrow$ | STL10  | CIFAR100  | Food101  | ImageNet |
> | :---- | :---: | :---: | :---: | :---: |
> | Last | 20.0 | 29.8 | 20.6 | 17.3 |
> | Similar | 17.9 | 29.6 | 20.5 | 17.9 |
> | Similar \+ R\_S | 9.4 | 33.2 | 27.8 | 25.5 |
>
> As the reviewer suggested, we further looked for classes that are entirely missclassified. Without the proposed framework, 520 out of the 1000 ImageNet classes were completely misclassified. When using the proposed framework, 168 classes that were previously entirely unrecognized now achieve non‑zero scores (with an average per-class F1 score of 18.6), while 35 new classes that have previously been partially correctly classified (with an average per-class F1 score of 3.9) are now misclassified.
>
> > 2\. Additional Ablations
>
> The intuition behind these specific choices is that we rely on the same distance function used within the standard contrastive objective, and by normalization, we remove any constant offset in the embedding space, which could otherwise dominate raw inner products or cosine similarities. Centering ensures that the subsequent similarity graphs reflect relative geometry.
>
> We further experimented with different normalization schemes and distance functions. Overall, our method demonstrates robustness to these changes, highlighting that the primary benefit stems from the idea of regularizing hierarchical relationships.
>
> |  | STL10 | CIFAR10 | Caltech101 | Food101 | CIFAR100 | ImageNet | Pets | Flickr I2T | Flickr T2I |
> | :---- | :---: | :---: | :---: | :---: | :---: | :---: | :---: | :---: | :---: |
> | Default setting | 92.6 | 96.3 | 56.0 | 30.6 | 51.3 | 24.7 | 13.2 | 65.8 | 53.7 |
> | Centering First | 92.1 | 96.4 | 55.8 | 30.8 | 51.1 | 24.7 | 13.7 | 65.9 | 53.4 |
> | No Centering | 92.7 | 96.1 | 55.9 | 30.5 | 51.5 | 24.7 | 13.4 | 66.1 | 53.7 |
> | Standard Scaling | 90.8 | 95.1 | 55.7 | 31.0 | 50.3 | 23.9 | 14.6 | 65.1 | 53.1 |
> | RBF Sim. | 90.8 | 94.9 | 55.7 | 30.9 | 50.3 | 23.9 | 14.3 | 65.0 | 53.0 |
> | Spearman Sim. | 90.4 | 91.6 | 55.2 | 30.0 | 50.0 | 23.3 | 13.9 | 63.6 | 51.9 |
>
> > 3\. Paired vs. Unpaired Samples for STRUCTURE
>
> We evaluated the influence of using paired or unpaired samples for computing STRUCTURE. Specifically, for all experiments, we keep the alignment dataset the same but we vary the samples on which we compute the regularization. In the default setting (called *paired*) we compute the regularization on 5k samples that are also used to train the alignment function. In the second experiment, suggested by the reviewer, we compute the regularization on 10k samples that are disjoint from the alignment dataset where 50% of samples are used to compute regularization for the vision model, and other 50% for the language model (called *unpaired*). Results show that computing structure on unpaired samples yields comparable benefits as using paired samples. This shows that the structural prior that STRUCTURE maintains is generalizable to the whole original space.
>
> |  | CIFAR10 | CIFAR100 | Caltech101 | Food101 | ImageNet |
> | :---- | :---: | :---: | :---: | :---: | :---: |
> | paired | 96.3 | 51.3 | 56.0 | 30.6 | 24.7 |
> | unpaired | 92.9 | 51.1 | 57.1 | 31.0 | 24.8 |
>
> > 4\. Fixed vs. Random Subset for STRUCTURE
>
> By default, for efficiency, STRUCTURE is computed on the current batch of samples, rather than on the entire dataset. We evaluated the influence of using a fixed same-size subset for computing STRUCTURE. Specifically, we compare the use of a fixed random sample from the dataset (fixed) with the standard approach of using the batches to compute the regularization term (random). Results show that the regularization yields very similar results regardless of whether it uses a fixed or a random subset of the data.
>
> |  | CIFAR10 | CIFAR100 | Caltech101 | Food101 | ImageNet |
> | :---- | :---: | :---: | :---: | :---: | :---: |
> | random | 96.3 | 51.3 | 56.0 | 30.6 | 24.7 |
> | fixed | 96.5 | 51.5 | 56.0 | 30.7 | 24.7 |
>
> ## Minor Weakness:
>
> > 1\. Additional Baseline: ASIF
>
> Unlike our method, ASIF \[1\] does multimodal matching *without learning*. We now include ASIF as a baseline, and the results demonstrate that our method substantially outperforms it, achieving relative improvements of 32.9% on the COCO retrieval task, 32.9% on the CIFAR-100 fine-grained classification task, and 5.6% on the CIFAR-10 coarse-grained classification task.
>
> |  | COCO | CIFAR 100 | CIFAR 10 |
> | :---- | :---: | :---: | :---: |
> | ASIF | 24.6 | 23.8 | 91.7 |
> | Ours | 32.7 | 46.9 | 96.8 |
>
> > 2\. Related Work
>
> We have revised the related work section to provide a broader context for our work within the field of representation alignment \[b, c, d, e\]. This updated section now positions the "Platonic representation hypothesis" within this more general framework.
>
> ## Minor comments:
>
> > 2\. Definition of Resource Constrained
>
> We use at least 1000 labeled samples in our experiments because, in the vision-language domain, this scale is often considered “resource-constrained” relative to the large datasets typically used. However, our method does not require 1000 labeled samples to be effective. For example, as shown in our response to Reviewer y6jJ (Q2), the results show that STRUCTURE remains beneficial with only 100 labeled samples in the biological domain.
>
> > 4\. Additional Experiment on “Train-test data distribution shift”
>
> We conducted additional experiment of “Train-test data distribution shift” in which we added a mix of 10 samples for each class in each dataset. Results show that this consistently improves performance across all datasets, rather than introducing any noise.
>
> |  | CIFAR100 | Pets | Flowers | Food101 | ImageNet |
> | :---- | :---: | :---: | :---: | :---: | :---: |
> | Target dataset only | 84.64 | 86.53 | 98.92 | 80.61 | 72.70 |
> | Mix all datasets | 84.04 | 86.75 | 99.54 | 80.74 | 72.61 |
>
> > 5\. Unifying Scale Up and Scale Down Experiment
>
> We did not unify the scaling up and down experiments because they rely on different data used to train the alignment functions. Specifically, for scaling down, we use COCO, and for scaling up, we add LAION samples to COCO to increase the sample size while maintaining the core as the same as in the other experiments. Furthermore, we wanted to keep them separate as they investigate different behaviours of the proposed method. Specifically, the scaling-down investigation examines how little data is needed to improve the standard approach using standard contrastive methods. In contrast, the scaling-up investigation explores where the influence of regularization diminishes when sufficient samples are available.
>
> > 6\. Similarity Distribution of Mutual kNN
>
> In general, for a given model pair, the most suitable layers are the layers with the highest mutual kNN score. However, we did not find a rule where these most suitable layers are found across different model pairs. Since the rebuttal policy forbids us from uploading images, we have added a table below that shows a subset of results. In particular, we aligned each of the last five layers from the vision and language encoder (RoBERTa and ViT-L) and evaluated their respective performance on the STL dataset. Results show that the most suitable layers are 24/22, and they perform significantly better than the others. Additionally, we report the alignment score in terms of mutual kNN in brackets for each combination as the similarity distribution. We will add comprehensive results across each layer as a heatmap in the revised manuscript.
>
> | Language Index vs. Vision Index | 20 | 21 | 22 | 23 | 24 |
> | :---- | :---: | :---: | :---: | :---: | :---: |
> | 19 | 88.2 (0.19) | 87.2 (0.18) | 86.8 (0.19) | 87.3 (0.20) | 88.6 (0.22) |
> | 20 | 84.4 (0.20) | 86.0 (0.19) | 82.3 (0.21) | 88.4 (0.22) | 91.7 (0.25) |
> | 21 | 80.5 (0.22) | 80.2 (0.21) | 93.6 (0.23) | 83.6 (0.24) | 93.4 (0.28) |
> | 22 | 95.1 (0.22) | 95.9 (0.21) | 95.2 (0.22) | 93.3 (0.24) | **97.3 (0.29)** |
> | 23 | 83.5 (0.21) | 83.4 (0.21) | 83.7 (0.22) | 83.4 (0.24) | 86.1 (0.28) |
>
> > 1&3&7\. Comments
>
> We have updated the manuscript in response to comments 1, 3, and 7\.

---

> > ### Comment · Reviewer_MNry · 2025-08-03
> >
> > I thank the authors for their work on the rebuttal. The extra content is great and thorough in addressing all of my doubts and questions, so I commend the authors on that!
> >
> > However, based on the evaluation criteria, I believe it doesn't fully meet the "groundbreaking" or "high impactful" requirement for a full 6 rating. I'm therefore happy to confirm my initial assessment of a strong submission with a 5.

---

> > > ### Author Response · Authors · 2025-08-04
> > >
> > > We thank the reviewer for the thoughtful and encouraging feedback. We're glad that our rebuttal addressed the questions thoroughly and that the reviewer views the submission as strong and comprehensive. We truly appreciate the engagement and the time the reviewer dedicated to evaluating our work.
> > >
> > > While we respect the reviewer’s assessment that grade 6 requires a “groundbreaking contribution”, we would like to highlight that our work demonstrates, for the first time, that high-quality multi-modal alignment is possible using 0.02% of the data typically employed in the field. This challenges prevailing assumptions about data requirements for multi-modal models and has the potential to significantly influence a broad range of multi-modal applications, especially in low-resource or domain-specific settings. In the rebuttal, we further extended our results beyond image-language and audio-language models by applying our approach to the biology domain, where we performed zero-shot classification of cell images and gene expression data. Finally, we would like to highlight that our work surprisingly shows that adding just a few in-domain samples can match the performance of large-scale multimodal models like CLIP trained on hundreds of millions of multimodal data.

---

### Decision · Program_Chairs · 2025-09-17

**Decision:**

Accept (poster)

**Comment:**

To address the multimodal alignment problem between unimodal foundational models, the authors proposed STRUCTURE, which enforces consistency between structural relationships (i.e., neighborhood structure) from the unimodal representations to the joint representation -- thus requiring *far* fewer aligned data pairs. Operationally, STRUCTURE can be applied to many existing alignment methods by adding a unimodal neighborhood preserving regularizer to the joint embeddings alignment procedure. Secondly, the authors empirically demonstrate (and theoretically argue) that selecting layers with a similarity-based criteria is better than jut selecting the last layer. Experiments are conducted for seven zero-shot classification benchmarks and two cross-modal retrieval benchmarks, demonstrating notably improved performance relative to the underlying approaches without STRUCTURE regularization (with additional baselines provided during rebuttal). Finally, secondary experiments are performed to quantify training data size requirements, distribution shift, validation of neighborhood preservation, sensitivity analyses, and a text-audio alignment task.

Strengths of this work identified by reviewers include:
- The paper is well-motivated, well-structured, and well-written -- I concur with the reviewers in that I was able to understand just about everything in one reading.
- Requiring less aligned training data to project unimodal representations into a joint embedding space is a potentially impactful finding and the proposed method does this in a simple way that applies to most any deep learning method.
- The set of experiments is over a large number of task/datasets with sufficient diversity and multi-modal alignment methods, showing consistently good performance. The reviewers all thought that the (primary and secondary) empirical results were strong.

Weaknesses of this work identified by reviewers include:
- There were several 'nit' clarification questions raised by reviewers (e.g., macro vs. micro averaging, additional ablations, paired vs. unpaired examples, more than two modalities, additional baselines, addition of another domain (biology), modality gap -- some requiring additional experiments.  However, they were well-handled during rebuttal such that the reviewers universally raised/maintained their scores. I am not sure the additional experiments were needed for the paper to be sufficiently convincing, but they definitely help. Obviously, these additional findings should be integrated into the next manuscript revision.

There was consensus from the reviewers that this is a well-motivated approach to an important (and widely studied) problem for which strong evidence of positive performance improvements is provided. Additionally, the paper is well-written and the rebuttal provided further convincing results to support the STRUCTURE approach to multi-modal alignment. I concur with the reviewer consensus and believe that many researchers in this space will adopt this method as it is widely applicable, conceptually appealing, and shown to perform well.